# Learning an Image Editing Model without Image Editing Pairs

**Nupur Kumari**[1]     **Sheng-Yu Wang**[1]     **Nanxuan Zhao**[2]     **Yotam Nitzan**[2]

**Yuheng Li**[2]     **Krishna Kumar Singh**[2]     **Richard Zhang**[2]

**Eli Shechtman**[2]     **Jun-Yan Zhu**[1]     **Xun Huang**[2]

[1]Carnegie Mellon University     [2]Adobe Research

## Abstract

Recent image editing models have achieved impressive results while following natural language editing instructions, but they rely on supervised fine-tuning with large datasets of input-target pairs. This is a critical bottleneck, as such naturally occurring pairs are hard to curate at scale. Current workarounds use synthetic training pairs that leverage the zero-shot capabilities of existing models. However, this can propagate and magnify the artifacts of the pretrained model into the final trained model. In this work, we present a new training paradigm that eliminates the need for paired data entirely. Our approach directly optimizes a few-step diffusion model by unrolling it during training and leveraging feedback from vision-language models (VLMs). For each input and editing instruction, the VLM evaluates if an edit follows the instruction and preserves unchanged content, providing direct gradients for end-to-end optimization. To ensure visual fidelity, we incorporate distribution matching loss (DMD), which constrains generated images to remain within the image manifold learned by pretrained models. We evaluate our method on standard benchmarks and include an extensive ablation study. Without any paired data, our method performs on par with various image editing diffusion models trained on extensive supervised paired data, under the few-step setting. Given the same VLM as the reward model, we also outperform RL-based techniques like Flow-GRPO.

## 1 Introduction

Large-scale text-to-image models have achieved remarkable success, generating images of high fidelity that closely align with textual descriptions Ramesh et al. (2022); Peebles & Xie (2023); Kang et al. (2023). Despite these advances, text-only conditioning offers limited user control and falls short for many downstream applications (Meng et al., 2022; Gal et al., 2023). In practice, users often wish to start with an existing image to perform tasks like adjusting local attributes, changing the style, or placing an object in a new context. These *image editing* operations require precise, image-guided control that text-only prompts cannot provide.

While collecting large-scale text–image pairs is relatively straightforward (Schuhmann et al., 2021), constructing supervised datasets for editing tasks is far more challenging. As one requires a *pair* of images, the input and its edited counterpart, along with the text instruction, and such data is rarely available online. Early methods addressed this by synthetically generating editing pairs (Brooks et al., 2023) from a pretrained model, using zero-shot editing techniques (Hertz et al., 2023). However, synthetic datasets can quickly become outdated with new and improved base models, and they risk amplifying and propagating artifacts of the synthetic editing process. More recent approaches extract frames from videos and annotate their differences (Chen et al., 2025; Song et al., 2023b; Krojer et al., 2024). Although promising, the applicability of this strategy is constrained by the diversity of transformations present in natural video sequences, where obtaining pixel-aligned before–and–after edited pairs is nearly impossible. A final alternative is to manually create training pairs (Winter et al., 2024; Magar et al., 2025), but this can be quite laborious and does not scale as easily.

In this work, we explore the possibility of training an image editing model *without any training pairs*. Our key idea is to leverage supervision from Vision Language Models (VLMs) (Liu et al., 2023a), relying on their general image-understanding capabilities to check whether the generated images satisfy the editing instructions. Prior works have studied the use of specialized models or general-purpose VLMs in improving generative models along dimensions such as text-alignment and aesthetic quality, primarily using reinforcement learning (Black et al., 2024; Liu et al., 2025a). In contrast, our method is the first to explore using gradient feedback from VLMs for general instruction-following, and we distill this feedback into a lightweight generative model that can generalize to arbitrary images and edit instructions. Our final method combines the VLM-feedback with a distribution matching loss to ensure that generated outputs remain in the realistic image domain while following the edit instructions. In summary, our contributions are threefold:

1. We propose NP-Edit (No-Pair Edit), a framework for training image editing models using gradient feedback from a Vision–Language Model (VLM), requiring *no paired supervision*.

2. For efficient training and effective VLM feedback, our formulation combines it with distribution matching loss to learn a *few-step* image editing model. The final model remains competitive with existing baselines trained on supervised data.

3. We conduct a comprehensive empirical study analyzing the impact of (i) different VLM backbones, (ii) dataset scale and diversity, and (iii) VLM loss formulation. Our findings show that performance improves directly with more powerful VLMs and larger datasets, demonstrating its strong potential and scalability.

## 2 RELATED WORKS

**Diffusion-based image editing.** Development of large-scale text-to-image models has enabled a wide range of downstream applications, including local image editing (Hertz et al., 2023; Meng et al., 2022), stylization (Sohn et al., 2023; Hertz et al., 2024; Jones et al., 2024), personalization and customization (Gal et al., 2023; Ruiz et al., 2023). These can broadly be viewed as different forms of image-editing capabilities. Early approaches often relied on zero-shot inference-time methods (Hertz et al., 2023; Parmar et al., 2023; Cao et al., 2023; Avrahami et al., 2023; Kim et al., 2023) or flexible but slow optimization-based techniques (Gal et al., 2023; Ruiz et al., 2023; Kumari et al., 2023).

To improve efficiency and robustness, subsequent works introduced training-based approaches (Brooks et al., 2023; Xiao et al., 2025; Chen et al., 2023; Fu et al., 2024; Sun et al., 2024). However, obtaining large datasets of image pairs remains challenging: synthetic curation (Brooks et al., 2023; Zhang et al., 2023; Zhao et al., 2024; Hui et al., 2024; Yang et al., 2024b; Tan et al., 2025; Cai et al., 2025; Kumari et al., 2025) risks becoming outdated as generative models improve, while human annotation (Winter et al., 2024; Magar et al., 2025; Ge et al., 2024; Sushko et al., 2025) is costly and labor-intensive. Recent efforts have explored constructing paired data from videos (Chen et al., 2025; Song et al., 2023b) or simulation environments (Yu et al., 2025), although these remain limited in either annotation diversity or visual realism. We also target similar image-editing capabilities but remove the need for paired data (Zhu et al., 2017), by using differentiable feedback from vision–language models instead of ground-truth edits.

**Post-training for image generation.** Post-training methods typically align image generators with human preferences using either Direct Preference Optimization (DPO) (Wallace et al., 2024; Yang et al., 2024a) or Reinforcement Learning (RL) (Black et al., 2024). While early RL-based works use feedback from a simple scalar reward model (Kirstain et al., 2023; Xu et al., 2023), the paradigm has recently been enhanced by employing sophisticated Vision-Language Models (VLMs) as "judges" to provide more generic and accurate reward signals (Ku et al., 2024).

Although post-training has been successfully applied to text-to-image generation, its use for image editing models has been less explored. Concurrently to our work, EARL (Ahmadi et al., 2025) begins to address this by using a VLM-as-a-judge framework to post-train an image-editing model with RL. However, RL-based approaches often depend heavily on good initialization, typically requiring a Supervised Fine-Tuning (SFT) phase with paired editing data. In contrast, our method leverages differentiable feedback from the VLM model, thereby obviating the need for an initial SFT stage and enables the learning of image editing models without the use of synthetically generated data.

Related to our work, Luo et al. (2025) recently introduced a method that incorporates gradient feedback from VLMs to satisfy various criteria, including the horizon line, style, and layout, in generated images. However, their framework operates in a per-example optimization setting, requiring costly LoRA fine-tuning (Hu et al., 2022) for each criterion and prompt pair, and also does not consider image editing tasks.

**Few-step diffusion models.** Standard diffusion (or flow-matching) models require many sampling steps to generate high-quality images. Many prior works reduce the number of denoising steps for faster sampling by predicting larger denoising steps, including consistency models (Kim et al., 2024; Geng et al., 2024; Song et al., 2023a; Yang et al., 2024c; Song & Dhariwal, 2024; Lu & Song, 2025; Heek et al., 2024; Frans et al., 2024; Zhou et al., 2025), shortcut models (Frans et al., 2024), meanflow (Geng et al., 2025), and inductive moment matching (Zhou et al., 2025). Another line distills a pre-trained multi-step teacher into a few-step student by matching ODE trajectories (Song et al., 2023a; Salimans & Ho, 2022; Geng et al., 2023), using adversarial loss (Sauer et al., 2024b; Kang et al., 2024; Yin et al., 2024a; Sauer et al., 2024a; Xu et al., 2024), or applying score distillation (Luo et al., 2023; Yin et al., 2024b;a; Zhou et al., 2024). In our framework, we adopt DMD (Yin et al., 2024b) as a distribution matching objective. This ensures that our few-step editing model's output remains in the real-image manifold defined by the pre-trained text-to-image teacher, while using VLM-feedback to ensure the model follows the editing instructions.

## 3 BACKGROUND

### 3.1 DIFFUSION MODELS

Diffusion or flow-based models are a class of generating models that learn the data distribution by denoising samples corrupted by different levels of Gaussian Noise (Ho et al., 2020; Song et al., 2021; Lipman et al., 2023). Given a real sample $\mathbf{x}$, a forward diffusion process creates noisy samples $\mathbf{x}^t = \alpha^t \mathbf{x} + \sigma^t \epsilon$ over time $t \in (0, 1]$, where $\epsilon \sim \mathcal{N}(\mathbf{0}, \boldsymbol{I})$, and $\alpha^t$, $\sigma^t$ define a noise schedule such that $\mathbf{x}^T \sim \mathcal{N}(\mathbf{0}, \boldsymbol{I})$ and $\mathbf{x}^0 = \mathbf{x}$. The denoising model is parameterized to reverse the forward diffusion process by either predicting the noise $\epsilon$ (Ho et al., 2020) added to the sample or velocity $\mathbf{v}$ towards the clean sample (Liu et al., 2023b; Salimans & Ho, 2022). In our work, we follow the flow-based formulation, with the forward denoising process being a linear interpolation, i.e., $\alpha^t = 1 - t$ and $\sigma^t = t$. The training objective for a flow-based model, with parameters $\theta$, can be simplified to the following:

$$\mathbb{E}_{\mathbf{x}^t, t, \mathbf{c}, \epsilon \sim \mathcal{N}(\mathbf{0}, \boldsymbol{I})} w_t ||\mathbf{v} - \mathbf{v}_\theta(\mathbf{x}^t, t, \mathbf{c})||, \tag{1}$$

where $\mathbf{v} = \epsilon - \mathbf{x}$ and $w_t$ is a time dependent weighting factor. The denoising network can be conditioned on other inputs $\mathbf{c}$, such as a text prompt, a reference image, or both, as in our case.

### 3.2 VISION LANGUAGE MODELS (VLMS)

Vision Language Models (VLMs) trained from multimodal image-text data have shown exemplary visual understanding and reasoning capabilities and can serve as a general-purpose visual model. A common strategy for training such large-scale VLMs is via visual instruction tuning (Liu et al., 2023a), which aligns a pre-trained Vision Encoder output with the input word embedding space of a pretrained Large Language Model (LLM). More specifically, the image $\mathbf{x}$ is encoded into a set of tokens using the vision encoder, $\mathbf{X}_v = g(\mathbf{x})$. The input question regarding the image and its ground truth answer are tokenized in the LLM input embedding space as $\mathbf{X}_q$ and $\mathbf{X}_a$, respectively. A projector module, $f_\phi$, projects the vision-encoded tokens into the LLM word embedding space and is trained via standard autoregressive loss to maximize the probability of predicting the correct answer:

$$p(\mathbf{X}_a | \mathbf{X}_v, \mathbf{X}_q) = \prod_{i=1}^{L} p\Big(a_i | f_\phi(\mathbf{X}_v), \mathbf{X}_q, \mathbf{X}_{a<i}\Big), \tag{2}$$

where $\mathbf{X}_a = [a_1 \cdots a_L]$ is of token length $L$, and $\mathbf{X}_{a<i}$ denotes all the tokens before the current prediction token index. The final loss simplifies to a cross-entropy over the total vocabulary length. In our experiments, we use LLaVa-OneVision-7B (Li et al., 2024) as the VLM that uses SigLIP (Zhai et al., 2023) vision encoder and Qwen-2 LLM (Qwen-Team, 2024), and is among the state-of-the-art VLMs of this scale.

## 4 METHOD

Given a pretrained text-to-image diffusion model $G_{\text{init}}$ and a dataset $\mathcal{X} = \{(\mathbf{y}_i, \mathbf{c}_i, \mathbf{c}_i^{\mathbf{y}}, \mathbf{c}_i^{\mathbf{x}})\}_{i=1}^N$ of reference image $\mathbf{y}$, corresponding edit instruction $\mathbf{c}$, and captions $\mathbf{c}^{\mathbf{y}}$ and $\mathbf{c}^{\mathbf{x}}$ that describe the reference and edited image respectively, we fine-tune $G_{\text{init}}$ into a *few-step* image editing model $G_\theta$ without requiring ground truth edited image $\mathbf{x}$ according to the edit instruction. Our approach, No-Pair (NP)-Edit, introduces a VLM-based loss to evaluate edit success and combines it with a distribution matching loss to ensure outputs remain within the natural image domain. Below, we first detail the construction of the dataset, then our training objective, and finally other implementation details.

### 4.1 EDIT INSTRUCTION DATASET

Each dataset sample consists of a real image $\mathbf{y}$ as reference and an associated edit instruction, $\mathbf{c}$. Following prior works (Liu et al., 2025b; Ye et al., 2025), we focus on several categories of local editing operations, such as *Add*, *Replace*, *Remove*, *Adjust shape*, *Action*, *Stylization*, *Text editing*, *Color*, *Material*, and *Background* change, as well as more free-form editing tasks such as *Customization* or *Personalization* (Gal et al., 2023; Ruiz et al., 2023; Kumari et al., 2023). Candidate instructions for each type are generated using Qwen2.5-32B VLM (Qwen-Team, 2025). Given an image-instruction pair, we further query the VLM to assess its validity and to suggest the caption, $\mathbf{c}^{\mathbf{x}}$, for the edited image. For the customization task, we restrict reference images to those showing a prominent central object, either filtered from a real image corpus or generated via the pretrained model (Tan et al., 2025; Kumari et al., 2025), and prompt the VLM to generate a caption that places the object in a novel background or context. In total, for local and free-form editing instructions, our dataset consists of $\sim 3M$ and $\sim 600K$ reference images, respectively. The input prompt to the Qwen2.5-32B VLM for each setup is shown in the Appendix D.

### 4.2 TRAINING OBJECTIVE

Training a diffusion or flow-based model (Ho et al., 2020; Liu et al., 2023b) for image editing without pairs presents a unique challenge. Standard diffusion training takes as input noised versions of a ground-truth image. In our setting, no such ground-truth edited image exists; thus, we cannot construct these intermediate noisy inputs. On the other hand, directly mapping noise to the edited image in a single step is naturally challenging and yields poor fidelity (see Appendix B). To address this, during training, we propose to unroll the backward diffusion trajectory starting from noise using a two-step sampling procedure (Song et al., 2023a). Specifically, given the reference image–instruction pair $(\mathbf{y}, \mathbf{c})$, the editing model $G_\theta$ first predicts a provisional clean image $\hat{\mathbf{x}}_\theta^0$ from noise $\epsilon$. Then, a second step refines this estimate by feeding an interpolated noisy input back into the model:

$$
\begin{aligned}
\hat{\mathbf{x}}_\theta^0 &= \epsilon - \hat{\mathbf{v}}_\theta, && \text{where } \hat{\mathbf{v}}_\theta \equiv G_\theta(\epsilon, t=1, \mathbf{c}, \mathbf{y}), \quad \epsilon \sim \mathcal{N}(\mathbf{0}, \boldsymbol{I}), \\
\mathbf{x}_\theta^0 &= \hat{\mathbf{x}}_\theta^t - t\mathbf{v}_\theta, && \text{where } \mathbf{v}_\theta \equiv G_\theta(\hat{\mathbf{x}}_\theta^t, t, \mathbf{c}, \mathbf{y}), \quad \hat{\mathbf{x}}_\theta^t = (1-t)\hat{\mathbf{x}}_\theta^0 + t\epsilon; \epsilon \sim \mathcal{N}(\mathbf{0}, \boldsymbol{I}), t \sim (0,1),
\end{aligned}
\tag{3}
$$

With the second step, the model is now trained on noisy intermediate states at timesteps determined by $t$, while being more efficient than a full backward unroll. In our method, we focus on *few-step* generation—specifically four steps—and restrict $t \in [t_1, t_2, t_3]$, a fixed schedule, in the second step. Few-step generator provides a better estimate of the denoised image, $\mathbf{x}_\theta^0$, at intermediate steps, which in turn enables effective VLM-based feedback. VLMs tend to give unreliable judgments when inputs are noisy or blurry (see Appendix B). This also enables faster inference and lowers training costs.

**VLM-based editing loss.** To evaluate whether an edit is successfully applied in $\mathbf{x}_\theta^0$, we define a set of template questions with corresponding ground truth answers, $\mathcal{D}_{\text{QA}} = \{(\mathbf{X}_{q_j}, \mathbf{X}_{a_j})\}_j$, tailored to each edit category. The VLM is instructed to answer with a binary yes and no answer, i.e., $\mathbf{X}_{a_j} \in \{Yes, No\}$, and $\mathbf{X}_{\bar{a}_j}$ denotes the opposite response. The loss is then a binary cross-entropy over the predicted logit difference between the tokens corresponding to the correct and opposite responses, respectively.

$$
\mathcal{L}_{\text{VLM}} = -\sum_j \log p(a_j), \text{ where } p(a_j) = \sigma(\ell_{a_j}^{(j)} - \ell_{\bar{a}_j}^{(j)})
\tag{4}
$$

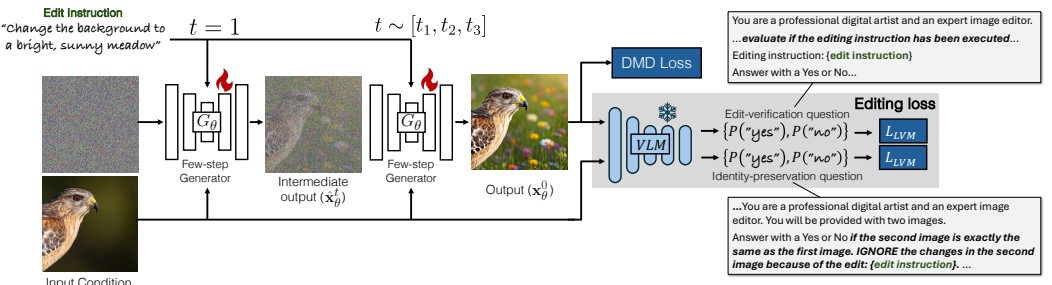

Figure 1: **Method.** We fine-tune a pretrained text-to-image model into a few-step image-editing model using differentiable VLM-feedback regarding edit success and distribution matching loss (DMD (Yin et al., 2024a)) for ensuring output images remain in the natural image manifold. Given the edit instruction and condition image, we predict the edited image starting from noise. During training, we randomly sample the few-shot timestep (Line 14 of Algorithm 1), and perform a two-step diffusion unrolling to predict the edited image for the intermediate timestep as shown here. The loss is backpropagated through the two diffusion sampling steps.

where $\ell_{a_j}^{(j)}$ is the logit corresponding to the token $\mathbf{X}_{a_j}$, $\sigma$ is the sigmoid function, and $p(a_j)$ is the probability of correct answer, while restricting normalization to only the *Yes* and *No* tokens, which we observe to be more effective during training (Zhang et al., 2024). Computing this loss is relatively fast, as it requires a single forward call to the VLM per question, as opposed to autoregressive prediction.

For each edit instruction, we use two complementary questions to compute the editing loss: (1) *Edit-verification* question to assess whether the intended edit is applied, and (2) *Identity-preservation* question to ensure the image is not over-edited and is consistent with the reference image. Specifically, for the local image-editing instructions, we verify edit success with the following question "The objective is to evaluate if the editing instruction has been executed in the second image. Editing instruction: {edit instruction}. Answer with a Yes or No." except *removal* edit-type, where we directly evaluate if the intended object is removed by asking "Answer with a Yes or No if the image has {object name}". For the identity-preservation question, we ask the following: "Answer with a Yes or No if the second image is exactly the same as the first image. IGNORE the changes in the second image because of the edit: {edit instruction}". We provide the list of all questions along with their system and user prompts for all editing types, including free-form editing in the Appendix E.

**Distribution matching with text-to-image teacher model.** While VLM feedback evaluates the efficacy of instruction following, it does not enforce the generated outputs to remain in the real image domain. To ensure this and keep the output distribution of the generator aligned with the pre-trained model, we apply Distribution Matching Distillation (DMD) (Yin et al., 2024b;a) between the fine-tuned model, $G_\theta$, and the pre-trained text-to-image (teacher) model, $G_{\text{init}}$. DMD minimizes the Kullback–Leibler (KL) divergence between the real image distribution, as estimated by the teacher model, and the output distribution of the fine-tuned model. The gradient of this KL-divergence loss with respect to the generator parameters can be simplified to:

$$\nabla_\theta D_{KL} = \mathbb{E}_{\epsilon \sim \mathcal{N}(\mathbf{0}, \mathbf{I}), t \in (0,1), \mathbf{x}_\theta^0} \left[ -\left( \mathbf{v}_{\text{real}}(\mathbf{x}_\theta^t, t, \mathbf{c}^{\mathbf{x}}) - \mathbf{v}_{\text{gen}}(\mathbf{x}_\theta^t, t, \mathbf{c}^{\mathbf{x}}) \right) \frac{dG}{d\theta} \right], \tag{5}$$

where $\mathbf{c}^{\mathbf{x}}$ is the text caption describing the noisy edited image $\mathbf{x}_\theta^t$ and $\mathbf{v}_{\text{real}}$, $\mathbf{v}_{\text{gen}}$ represents the predicted velocity from the teacher and a trainable auxiliary model, $A_\phi$ respectively. The auxiliary model is trained along with $G_\theta$ to learn the current output distribution of $G_\theta$ using a flow-based denoising objective. This loss ensures that the edited images not only satisfy the instruction but also remain faithful to the text-conditioned distribution of real images modeled by the pretrained teacher.

### 4.3 TRAINING DETAILS

The pretrained model $G_\theta$ is originally designed to generate an image, $\mathbf{x}$, conditioned only on text $\mathbf{c}$. To adapt it to our editing task, we extend its conditioning to include the reference image $\mathbf{y}$. Following recent works (Xiao et al., 2025; Tan et al., 2025), we concatenate the VAE encoding of the reference image to the noisy target image encoding along the token sequence dimension, similar to text embedding, thereby enabling the model to attend to both text and visual conditions.

To stabilize training, in the initial few iterations, we train the model with the objective of simply reconstructing the concatenated reference image. This encourages the network to propagate content

from the reference input, aligning it toward producing realistic images under joint text–image conditioning. After this, we introduce our main training objective as explained in the previous section.

The final loss for the generator is a weighted combination of the VLM-based editing loss and DMD loss. The auxiliary network, $A_\phi$, is updated $N_{\text{aux}}$ times for every generator, $G_\theta$, update (Yin et al., 2024a). Our pre-trained generative model is a 2B parameter internal DiT-based (Peebles & Xie, 2023) latent space diffusion model. The overall training pipeline is illustrated in Figure 1 and is detailed more formally in Algorithm 1 below. Other training hyperparameters are detailed in Appendix E.

---

**Algorithm 1:** NP-Edit: our training method

**Input:** Pretrained VLM and text-to-image model $G_{\text{init}}$, Dataset $\mathcal{X} = \{(\mathbf{y}_i, \mathbf{c}_i, \mathbf{c}_i^{\mathbf{y}}, \mathbf{c}_i^{\mathbf{x}})\}$.
**Output:** Few-step image-editing model $G_\theta$

1  $G_\theta \leftarrow \text{copyWeights}(G_{\text{init}}); \quad A_\phi \leftarrow \text{copyWeights}(G_{\text{init}})$
2  // Warmup with identity loss
3  **for** $step = 1$ **to** $N_{warmup}$ **do**
4      $(\mathbf{y}, \mathbf{c}^{\mathbf{y}}) \sim \mathcal{X}, \epsilon \sim \mathcal{N}(\mathbf{0}, \mathbf{I}), t \sim (0,1]$
5      $\mathbf{x} \leftarrow \mathbf{y}$
6      $\mathbf{x}^t \leftarrow (1-t)\mathbf{x} + t\epsilon$
7      $\mathbf{v}_\theta \leftarrow G_\theta(\mathbf{x}^t, t, \mathbf{y}, \mathbf{c}^{\mathbf{y}})$
8      $\mathcal{L}_{\text{id}} \leftarrow \|\mathbf{v} - \mathbf{v}_\theta\|$ where $\mathbf{v} = \epsilon - \mathbf{x}$
9      $\theta_G \leftarrow \theta_G - \eta_G \nabla_{\theta_G} \mathcal{L}_{\text{id}}.$
10  **end for**
11
12  // Main training loop
13  **while** $train$ **do**
14      $\{(\mathbf{y}, \mathbf{c}, \mathbf{c}^{\mathbf{x}})\} \sim \mathcal{X}, \epsilon \sim \mathcal{N}(0, I), t \in [t_1, t_2, t_3, t_4 = 1]$
15      $\mathbf{v}_\theta \leftarrow G_\theta(\epsilon, t = 1, \mathbf{y}, \mathbf{c})$
16      $\mathbf{x}_\theta^0 \leftarrow \epsilon - \mathbf{v}_\theta$
17      **if** $t < 1$ **then**
18          $\mathbf{v}_\theta \leftarrow G_\theta(\mathbf{x}_\theta^t, t, \mathbf{y}, \mathbf{c}); \quad \mathbf{x}_\theta^t \leftarrow (1-t)\mathbf{x}_\theta^0 + t\epsilon ; \quad \epsilon \sim \mathcal{N}(\mathbf{0}, \mathbf{I})$
19          $\mathbf{x}_\theta^0 \leftarrow \mathbf{x}_\theta^t - t\mathbf{v}_\theta$
20      **end if**
21      Compute $\mathcal{L}_{\text{VLM}}$ // Eqn. 4
22      Compute $\nabla_\theta D_{KL}$ // Eqn. 5
23      $\theta_G \leftarrow \theta_G - \eta_G \lambda_{\text{vlm}} \nabla_{\theta_G} \mathcal{L}_{\text{VLM}} - \eta_G \lambda_{\text{dmd}} \nabla_\theta D_{KL}$
24      **for** $local\ step = 1$ **to** $N_{aux}$ **do**
25          $\epsilon \sim \mathcal{N}(\mathbf{0}, \mathbf{I}), \{(\mathbf{y}, \mathbf{c}, \mathbf{c}^{\mathbf{x}})\} \sim \mathcal{X}$
26          $\mathbf{x}_\theta^0 \leftarrow G_\theta(\epsilon, \mathbf{y}, \mathbf{c})$ // edited image with backward unroll
27          $\mathbf{x}_\theta^t \leftarrow (1-t)\mathbf{x}_\theta^0 + t\epsilon \quad \epsilon \sim \mathcal{N}(\mathbf{0}, \mathbf{I}) \ t \in (0,1)$
28          $\mathbf{v}_\phi \leftarrow A_\phi(\mathbf{x}_\theta^t, \mathbf{c}^{\mathbf{x}})$
29          $\phi_A \leftarrow \phi_A - \eta_A \nabla_\theta \|\mathbf{v}_\phi - \mathbf{v}\|$ where $\mathbf{v} = \epsilon - \mathbf{x}_\theta^0$
30      **end for**
31  **end while**

---

## 5 EXPERIMENTS

In this section, we show the results of our method on local image editing as well as more free-form image editing tasks like customization, and compare them with the state-of-the-art baseline methods.

### 5.1 LOCAL IMAGE-EDITING

**Benchmark.** For evaluation, following prior works, we use the English subset of GEdit-Benchmark Liu et al. (2025b), which captures real-world user interactions across different edit types. We also show results on the ImgEdit (Ye et al., 2025) benchmark in Appendix A.

**Evaluation metric.** For quantitative evaluation, we follow prior works and use GPT4o-based VIEScore (Ku et al., 2024) metric. It scores each edit on: (1) Semantic Consistency (SC) score,

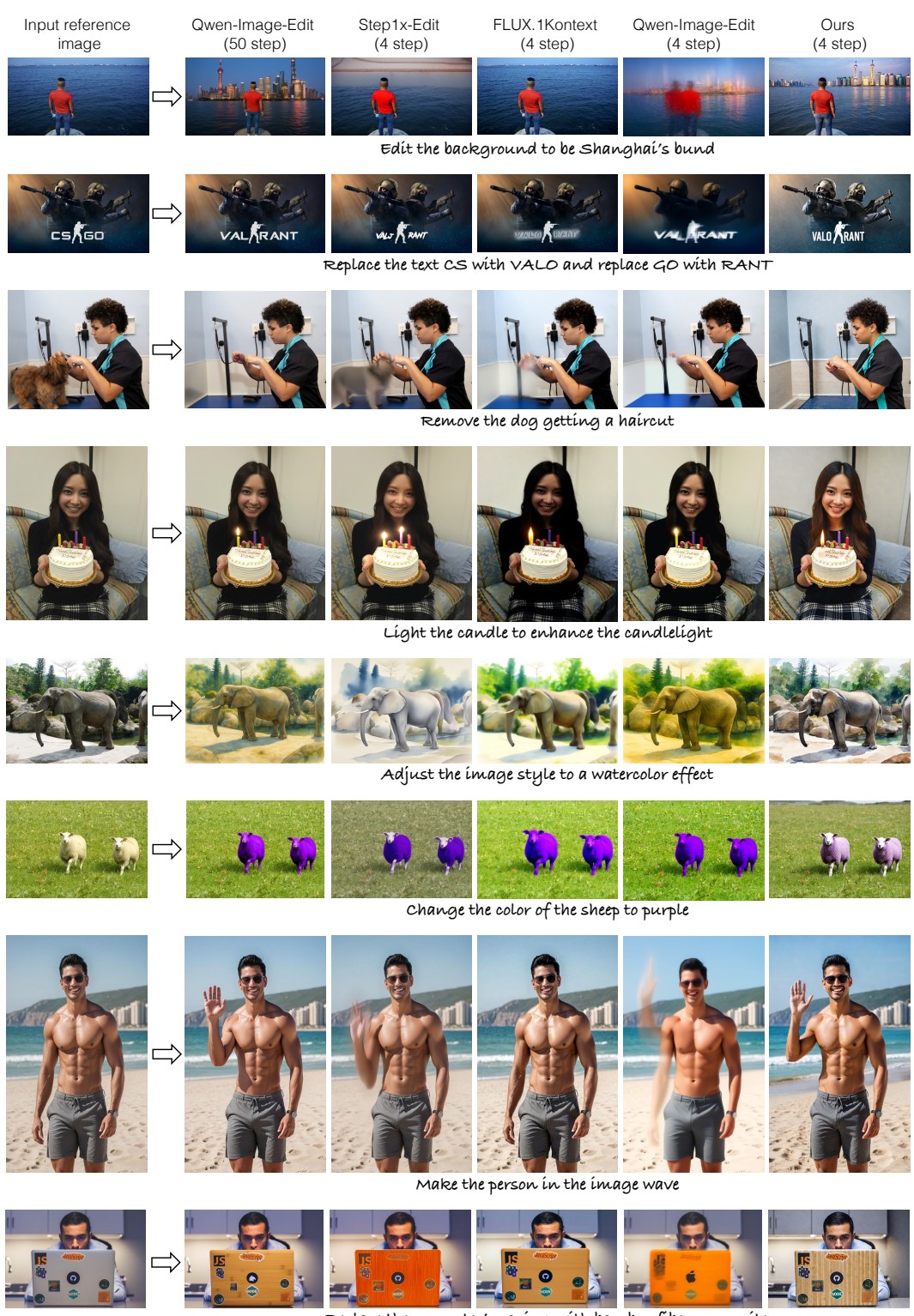

Figure 2: **Qualitative comparison on GEdit-Bench** under the few-step sampling setting. For an upper-bound comparison, in the 1$^{st}$ column we show results of the best multi-step sampling method (as measured by the quantitative metrics in Table 1). Our method performs on par or better than baseline methods across different edit types in the few-step setting. We show more samples in the Appendix Figure 13

Table 1: **Quantitative evaluation on GEdit-Bench**. Our method performs on par or better than baselines under the few-step setting. For multi-step sampling, it still outperforms OmiGen and remains competitive with many of the larger-scale models like BAGEL and FLUX.1 Kontext. All numbers reported in $\times 10$

| Method | #Param | #Step | SC Score↑ | PQ Score ↑ | Overall ↑ |
|---|---|---|---|---|---|
| Omni-Gen (Xiao et al., 2025) | 4B | 50 | 5.52 | 6.14 | 4.97 |
| BAGEL (Deng et al., 2025) | 7B | 50 | 7.02 | 6.26 | 6.14 |
| FLUX.1-Kontext (Labs et al., 2025) | 12B | 28 | 6.29 | 6.65 | 5.65 |
| Step-1X Edit (Liu et al., 2025b) v1.1 | 12B | 28 | 7.30 | 7.37 | 6.79 |
| Qwen-Image-Edit (Wu et al., 2025) | 20B | 50 | **7.94** | **7.50** | **7.36** |
| FLUX.1-Kontext (Labs et al., 2025) | 12B | 4 | 5.80 | 5.74 | 5.04 |
| Step-1X Edit (Liu et al., 2025b) v1.1 | 12B | 4 | 6.61 | 6.43 | 6.01 |
| Qwen-Image-Edit (Wu et al., 2025) | 20B | 4 | **6.82** | 6.21 | 6.06 |
| Turbo-Edit (Deutch et al., 2024) | 1B | 4 | 3.84 | 6.67 | 3.84 |
| NP-Edit (Ours) | 2B | 4 | 6.16 | **7.69** | **6.10** |

Table 2: **Free-form editing task, *Customization*, evaluation on Dreambooth**. We perform better than OminiControl, DSD, and SynCD, which are trained for this task on synthetic datasets. When compared to FLUX.1-Kontext and Qwen-Image-Edit, we still perform comparably in the few-step setting. All numbers are reported in $\times 10$

| Method | #Param | #Step | SC Score↑ | PQ Score ↑ | Overall ↑ |
|---|---|---|---|---|---|
| DSD (Cai et al., 2025) | 12B | 28 | 6.71 | 7.41 | 6.78 |
| SynCD (Kumari et al., 2025) | 12B | 30 | 7.66 | 7.83 | 7.54 |
| FLUX.1-Kontext (Labs et al., 2025) | 12B | 28 | 8.19 | 7.45 | 7.61 |
| Qwen-Image-Edit (Wu et al., 2025) | 20B | 50 | **8.53** | **7.79** | **8.02** |
| OminiControl (Tan et al., 2025) | 12B | 8 | 6.33 | **7.82** | 6.22 |
| DSD (Cai et al., 2025) | 12B | 8 | 6.37 | 6.78 | 6.29 |
| SynCD (Kumari et al., 2025) | 12B | 8 | 7.71 | 6.84 | 7.07 |
| FLUX.1-Kontext (Labs et al., 2025) | 12B | 8 | 7.99 | 7.18 | 7.39 |
| Qwen-Image-Edit (Wu et al., 2025) | 20B | 8 | **8.08** | 7.44 | **7.62** |
| NP-Edit (Ours) | 2B | 8 | 7.68 | 7.56 | 7.33 |
| NP-Edit (Ours) | 2B | 4 | 7.60 | 7.28 | 7.10 |

evaluating whether the edit instruction was followed, and (2) Perceptual Quality (PQ) score, assessing realism and absence of artifacts. Following VIEScore, for the *Overall score*, we take the geometric mean between SC and PQ for each image, and average across images in the evaluation benchmark.

**Baselines.** We compare our method with leading baselines, including FLUX.1-Kontext (Labs et al., 2025), Step1X-Edit (Liu et al., 2025b), BAGEL (Deng et al., 2025), OmniGen (Xiao et al., 2025), and Qwen-Image-Edit (Wu et al., 2025). Since no prior work explicitly targets few-step editing, we simply evaluate the above baselines with few-step sampling as well as their original multi-step setting for an upper-bound comparison. We also include Turbo-Edit (Deutch et al., 2024), a state-of-the-art zero-shot few-step method that requires no paired supervision (as zero-shot) and is thus closest to our setup. We use the open-source implementation of all baselines, with further details in the Appendix F.

**Results** Table 1 shows the quantitative result. In the few-step setting, our method achieves the best Overall and Perceptual Quality (PQ) score compared to baseline methods. When compared to their original multi-step sampling, our few-step model still outperforms OmniGen and remains competitive with BAGEL, FLUX.1-Kontext, despite being $\times 6$ smaller parameter-wise. While Step1X-Edit and Qwen-Image-Edit perform better, they are substantially larger models. Figure 2 provides a qualitative comparison. As we can see, our method can successfully follow different editing instructions while being consistent with the input reference image. For instance, in the 6[th] row (sheep color change), our approach produces a more natural edit compared to baselines. It also performs comparably to the multi-step variant for edits like lighting the candle in 4[rth] row or making the person wave in 7[th] row.

## 5.2 FREE-FORM EDITING: CUSTOMIZATION

**Benchmark.** We use the widely adopted DreamBooth (Ruiz et al., 2023) dataset for evaluation. It consists of 30 objects and 25 prompts per object category. The goal is to generate the same object as shown in the reference image, but in a different context, as mentioned in the text prompt.

**Baselines.** We compare against state-of-the-art unified image-editing baselines such as FLUX.1-Kontext (Labs et al., 2025) and Qwen-Image-Edit Wu et al. (2025) as well as OminiControl (Tan

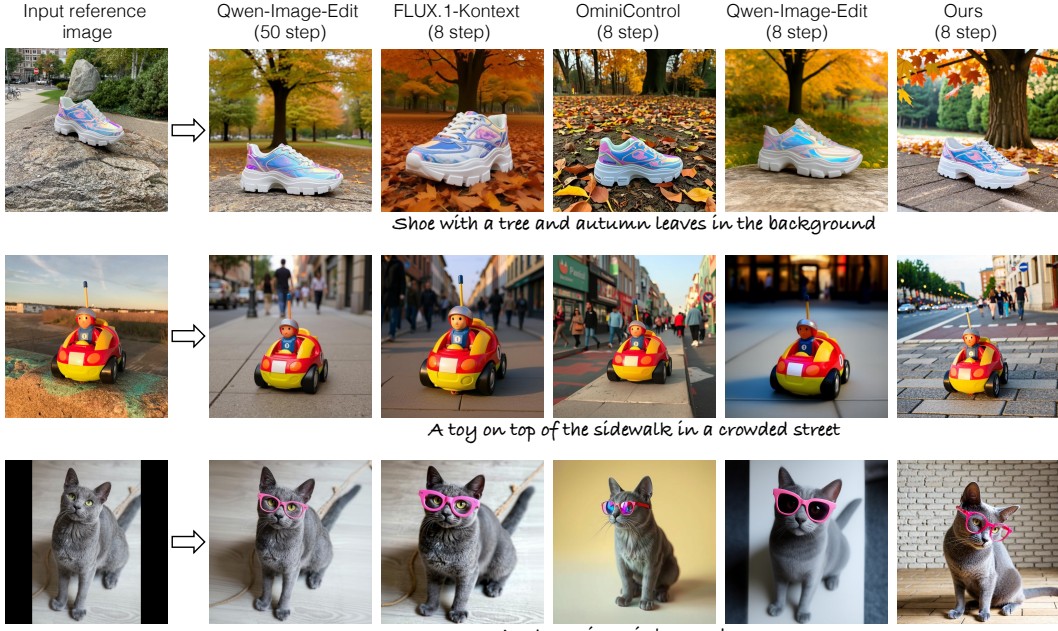

Figure 3: **Qualitative comparison on Customization task.** Our method can generate the object in new contexts while having better fidelity under few-step sampling. We show more samples in the Appendix Figure 15.

et al., 2025), DSD (Cai et al., 2025), and SynCD (Kumari et al., 2025), which are feed-forward models trained specifically for this task on synthetic datasets.

**Evaluation metric.** Here as well, we use the VIEScore evaluation with a similar Semantic Consistency (SC) score to evaluate identity and text alignment, a Perceptual Quality (PQ) score to measure realism, and the geometric mean of the two for the Overall score. We also report CLIPScore (Radford et al., 2021) and DINO (Oquab et al., 2023) similarity-based metrics in the Appendix.

**Results.** As shown in Table 2, our method performs comparably to state-of-the-art methods. In the few-shot sampling setting, all the baseline methods fail to generate realistic samples at 4 steps; therefore, we compare with them at 8 sampling steps. Our method still results in higher fidelity samples as Figure 3 shows, while maintaining object identity with the reference image. Note that our method performs better than OminiControl, which is also a few-step (8) model for this task.

### 5.3 ABLATION

In this section, we perform several ablations to analyze the role of different components of our method, dataset scale, and stronger VLMs. All ablations are done on the local image-editing task.

**Training objective.** We ablate our training objective across four settings: (1) using only distribution matching loss, (2) using only the VLM-based editing loss, (3) removing the identity-preservation question from $\mathcal{D}_{QA}$, and (4) replacing the binary-cross entropy loss (Eqn. 4) with standard cross-entropy over the full vocabulary. Results are shown below in Table 3. Training without the VLM-based loss and relying solely on distribution matching significantly degrades the model's capabilities at following editing instructions. We observe that VLM-based loss is essential for maintaining consistency between input and edited images and for certain editing tasks like *Removal* (Figure 4 and Appendix Figure 5). However, only training with VLM-based loss leads to unrealistic outputs (Appendix Figure 6), and the training eventually diverges, as evidenced by the low overall score in Table 3, underscoring the need for DMD loss. In addition, using binary cross-entropy loss and having a question to check consistency between input and edited images improves the overall performance.

**Dataset and VLM scale.** To study the role of dataset scale, we vary the number of unique reference images in training. Our final dataset represents the maximum scale feasible under our computational resources. Table 4 shows the performance across different dataset sizes and VLM backbones. We observe consistent gains with larger datasets, suggesting that further scaling of data could yield additional improvements. Similarly, a larger parameter VLM-backbone leads to better performance,

Table 3: **Training objective ablation**. We compare on the GEdit-Bench using the VIEScore metric. Ablating different components of our method leads to a drop in performance, indicating its importance.

| Method | SC Score↑ | PQ Score ↑ | Overall ↑ |
|---|---|---|---|
| Ours | **6.16** | **7.69** | **6.10** |
| w/ only DMD | 4.93 | 7.51 | 4.93 |
| w/ only VLM | 2.03 | 3.48 | 1.93 |
| w/o VLM identity | 5.70 | 7.67 | 5.76 |
| w/ standard CE loss | 5.95 | 7.64 | 5.89 |

Table 4: **Dataset and VLM scale and comparison with Reinforcement Learning** on the GEdit-Bench. Increasing dataset scale and using stronger VLMs leads to increased performance. Our method also performs better than post-training an SFT model with RL (Liu et al., 2025a).

| Method | SC Score↑ | PQ Score ↑ | Overall ↑ |
|---|---|---|---|
| 1 % Dataset | 4.41 | 7.10 | 4.66 |
| 50 % Dataset | 5.41 | **7.73** | 5.52 |
| 100 % Dataset | **6.16** | 7.69 | **6.10** |
| InternVL-2B | 5.36 | 7.67 | 5.45 |
| InternVL-14B | 5.88 | 7.74 | 5.89 |
| LLava-0.5B | 4.57 | 7.50 | 4.59 |
| LLava-7B (Ours) | **6.16** | **7.69** | **6.10** |
| SFT | 3.91 | 5.70 | 3.64 |
| SFT + RL | 4.55 | 5.47 | 4.19 |
| SFT + Ours | **6.08** | **7.83** | **6.06** |

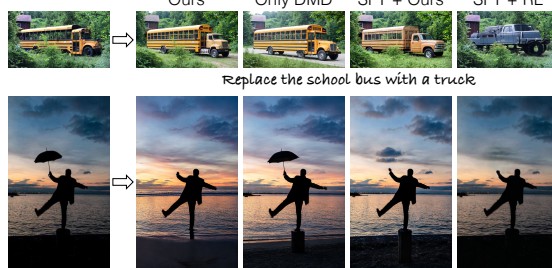

Figure 4: **Qualitative analysis of ablation experiments.** Our method maintains better input and edited image alignment compared to only training with DMD loss, which also fails on tasks like removal. Compared to fine-tuning an SFT model with RL, our method results in better fidelity while following the edit instruction. Please zoom in for details.

across different VLMs such as InternVL (Chen et al., 2024) and LLava-OneVision (Li et al., 2024), underscoring the promise that our method can improve as more powerful VLMs are developed.

**Our method vs. Reinforcement Learning (RL).** RL is a common post-training strategy for improving pre-trained models without paired supervision and can also leverage VLMs as the reward model, a similar setup to ours. Thus, we benchmark our method against Flow-GRPO (Liu et al., 2025a), a widely used RL method for text-to-image diffusion. However, since RL relies on a reasonable initialization, we need to first train an image-editing model via Supervised Fine-Tuning (SFT) on a paired dataset (Lin et al., 2025). We then fine-tune it with Flow-GRPO using the same Llava-OneVision reward model as in our approach. As shown in Table 4, SFT alone performs poorly, likely due to the limited quality of paired data. Our method surpasses both SFT and SFT+RL, despite requiring no paired supervision. Fine-tuning the model with some paired data before applying our approach can slightly improve the pixel-level consistency between the input reference and output edited image (as shown in Figure 4), although the quantitative numbers are similar.

The Appendix provides additional results and a more detailed discussion of the method's limitations.

## 6  DISCUSSION AND LIMITATIONS

This paper introduces a new paradigm for enabling image editing capabilities given a pre-trained text-to-image diffusion model, without paired before-and-after edit supervision. Our approach combines differentiable feedback from VLMs to ensure editing success with a distribution matching objective to maintain visual realism. This method achieves competitive performance with recent state-of-the-art baselines trained on paired data while enabling efficient few-step generation.

Despite these promising results, our method has limitations. Without pixel-level supervision, edits may deviate from the input image in fine-grained details or fail to fully preserve subject identity. We show in Appendix C that adding a perceptual similarity loss (e.g., LPIPS (Zhang et al., 2018)) between input and edited images alleviates this to some extent, though often at the cost of editing quality. Another concern for our method is that it is directly tied to the VLM's capabilities and its biases. Additionally, the requirement to keep it in GPU memory introduces VRAM overhead. We expect ongoing advances in stronger and more efficient VLMs can help address this issue. Overall, our framework scales effectively with large unpaired datasets and highlights a path toward more flexible post-training of generative models for diverse downstream tasks.

**Acknowledgment.** We thank Gaurav Parmar, Maxwell Jones, and Ruihan Gao for their feedback and helpful discussions. This work was partly done while Nupur Kumari was interning at Adobe Research. Sheng-Yu Wang is supported by the Google PhD Fellowship. The project was partly supported by Adobe Inc., the Packard Fellowship, the IITP grant funded by the Korean Government (MSIT) (No. RS-2024-00457882, National AI Research Lab Project), NSF IIS-2239076, and NSF ISS-2403303.

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

# Appendix

## A    ADDITIONAL COMPARISON WITH BASELINE METHODS

**Human preference study.**   We perform a pairwise human comparison study on GEdit-Bench using Amazon Mechanical Turk. To measure editing success (similar to SC score in VIEScore), following Imagic (Kawar et al., 2023), we show the reference image in the middle and the images edited by our method and a baseline method, on the left and right randomly, and ask the user: "Which of the two images (Left or Right) matches with the editing instruction better, while preserving the details from the input reference image shown in the middle?". We collect 500 responses per pairwise comparison and show the results in Table 5. Under the few-step setting, our method outperforms Turbo-Edit, marginally preferred over FLUX.1-Kontext, Step-1X Edit, and has a random chance when compared with Qwen-Image-Edit. This is consistent with the VIEScore evaluation in Table 1. Similar to editing success, to measure perceptual quality (similar to PQ score in VIEScore), we show the images edited by our method and a baseline method on the left and right randomly, and ask the user: "Which image do you think has better overall quality and photorealism?". As Table 5 shows, here our method is preferred over all the baselines, also consistent with the VIEScore evaluation.

**Local image-editing.**   Here, we compare on another commonly adopted image-editing benchmark, ImgEdit (Ye et al., 2025) (Basic), which covers nine local editing types across diverse semantic categories with a total of 734 samples. Quantitative results under their proposed GPT4o-based evaluation protocol are reported in Table 6, with VIEScore (Ku et al., 2024) results in Table 7. Consistent with the trend observed on GEdit-Bench, our method has better or on-par performance in the few-step setting and remains comparable to many of the multi-step baselines as well. We also include a comparison on TEDBench (Kawar et al., 2023), which focuses primarily on non-rigid and background edit-types. Table 8 shows the quantitative comparison with similar trends as GEdit-Bench and ImgEdit (Basic). Qualitative comparisons are shown in Figure 14 and 16 for ImgEdit (Basic) and TEDBench, respectively. We also show additional qualitative samples on GEdit-Bench in Figure 13.

**Customization or free-form editing.**   Here, we report additional metrics commonly used for evaluation. Specifically, CLIPScore (Radford et al., 2021) and TIFA (Hu et al., 2023) to measure text alignment; and similarity in DINOv2 (Oquab et al., 2023) feature space after background masking to measure identity alignment, denoted as MDINOv2-I. We also report an overall Geometric score (Yan et al., 2023) by taking the geometric mean of TIFA and MDINOv2-I. The results are shown in

Table 9. Consistent with the VIEScore evaluation reported in the main paper, our method performs comparably with other baselines in the few-step sampling regime. We show more sample comparisons in Figure 15.

fal

Table 5: **Human evaluation on GEdit-Bench** under few-step sampling setting. We report the percentage of users who prefer our method over each baseline. For editing success, we outperform Turbo-Edit and remain competitive with other baselines, with a slight preference over FLUX.1-Kontext and Step-1X Edit and random chance with Qwen-Image-Edit. This trend is consistent with the VIEScore evaluation in Table 1. The standard error for all is within $\pm 4\%$.

| **Ours** vs | **Turbo-Edit** | **FLUX.1-Kontext** | **Step-1X Edit** | **Qwen-Image-Edit** |
|---|---|---|---|---|
| SC preference | 63.19 % | 53.35 % | 56.46 % | 48.44 % |
| PQ preference | 73.92 % | 79.91 % | 80.39 % | 71.98 % |

Table 6: **ImgEdit-Bench comparison** using their proposed GPT-4o based evaluation protocal and few-step sampling setting. Our method outperforms baseline methods on the *Avg* of all edit types.

| Method | #Param | Action↑ | Bg ↑ | Style ↑ | Adjust ↑ | Replace ↑ | Add ↑ | Extract ↑ | Remove ↑ | Compose ↑ | Avg ↑ |
|---|---|---|---|---|---|---|---|---|---|---|---|
| Qwen-Image-Edit | 20B | 3.14 | 2.83 | 3.70 | 3.25 | 3.00 | 3.52 | 1.96 | 2.71 | 3.06 | 3.02 |
| Flux.1-Kontext | 12B | 3.51 | 2.97 | 3.89 | 3.04 | 3.15 | 3.31 | 1.82 | 2.37 | 2.46 | 2.95 |
| Step1X Edit | 12B | 3.66 | 2.60 | 3.46 | 3.44 | 2.50 | 3.25 | 1.77 | 2.41 | 2.38 | 2.83 |
| NP-Edit (Ours) | 2B | 4.44 | 4.13 | 4.14 | 3.94 | 3.57 | 4.52 | 2.01 | 2.71 | 3.18 | **3.63** |

Table 7: **VIEScore evaluation on ImgEdit-Bench**. Our method performs on par or better than baselines under the few-step setting. For multi-step sampling, it still outperforms OmniGen and remains competitive with many of the larger-scale models like BAGEL and FLUX.1 Kontext. All numbers reported in $\times 10$

| Method | #Param | #Step | SC Score↑ | PQ Score ↑ | Overall ↑ |
|---|---|---|---|---|---|
| BAGEL | 7B | 50 | 7.55 | 6.22 | 6.47 |
| FLUX.1-Kontext | 12B | 28 | 6.94 | 6.73 | 6.19 |
| Step-1X Edit v1.1 | 12B | 28 | 7.26 | 7.30 | 6.72 |
| QwenImage Edit | 20B | 50 | **8.30** | **7.77** | **7.85** |
| QwenImage Edit | 20B | 4 | 6.23 | 5.14 | 5.46 |
| FLUX.1-Kontext | 12B | 4 | 6.08 | 5.22 | 5.14 |
| Step-1X Edit v1.1 | 12B | 4 | 6.00 | 5.37 | 5.14 |
| NP-Edit (Ours) | 2B | 4 | **6.72** | **7.78** | **6.62** |

Table 8: **VIEScore evaluation on TEDBench**. Our method performs on par or better than baselines under the few-step setting and remains competitive to baselines with multi-step sampling. All numbers reported in $\times 10$

| Method | #Param | #Step | SC Score↑ | PQ Score ↑ | Overall ↑ |
|---|---|---|---|---|---|
| FLUX.1-Kontext | 12B | 28 | 6.54 | 6.77 | 5.97 |
| Step-1X Edit v1.1 | 12B | 28 | 6.86 | 7.62 | 6.67 |
| QwenImage Edit | 20B | 50 | **7.90** | **7.88** | **7.58** |
| QwenImage Edit | 20B | 4 | 6.20 | 5.32 | 5.44 |
| FLUX.1-Kontext | 12B | 4 | 5.84 | 4.96 | 4.92 |
| Step-1X Edit v1.1 | 12B | 4 | 6.24 | 5.90 | 5.39 |
| NP-Edit (Ours) | 2B | 4 | **6.96** | **8.28** | **6.98** |

# B ABLATION STUDY

**Training objective.**   Here, we provide a more detailed analysis by examining performance across different editing sub-types. As a reminder, we ablated our training objective under four settings: (1) using only the distribution matching loss, (2) using only the VLM-based editing loss, (3) removing the identity-preservation question from $\mathcal{D}_{QA}$, and (4) replacing the binary-cross entropy loss as explained in Eqn. 4 with standard cross-entropy over full vocabulary length. As shown in Figure 5, training with only the DMD loss yields comparable performance on certain sub-edit types such as

Table 9: **Quantitative evaluation of free-form editing task, *Customization*, on DreamBooth dataset**. All numbers reported in $\times 10$

| Method | #Param | #Step | MDINO Score↑ | CLIP Score ↑ | TIFA ↑ | Geometric Score ↑ |
|---|---|---|---|---|---|---|
| DSD | 12B | 28 | 6.55 | 3.08 | 8.71 | 7.32 |
| SynCD | 12B | 30 | 7.34 | 3.09 | 8.53 | 7.71 |
| FLUX.1-Kontext | 12B | 28 | 7.72 | 3.07 | 8.88 | 8.14 |
| Qwen-Image-Edit | 20B | 50 | **7.47** | **3.14** | **9.37** | **8.22** |
| OminiControl | 12B | 8 | 6.16 | 3.02 | 8.12 | 6.64 |
| DSD | 12B | 8 | 5.88 | 3.15 | 8.93 | 6.99 |
| SynCD | 12B | 8 | 7.11 | **3.16** | 9.11 | 7.79 |
| FLUX.1-Kontext | 12B | 8 | **7.50** | 3.08 | 8.83 | **7.98** |
| Qwen-Image-Edit | 20B | 8 | 7.29 | 3.08 | **8.96** | 7.91 |
| NP-Edit (Ours) | 2B | 8 | 6.82 | 2.97 | 8.73 | 7.54 |
| NP-Edit (Ours) | 2B | 4 | 7.03 | 3.04 | 8.89 | 7.72 |

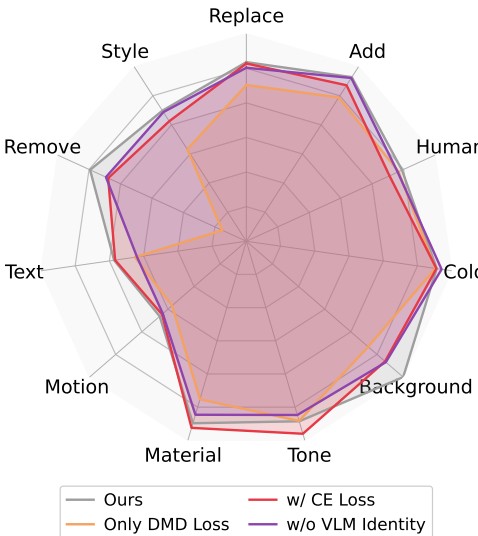

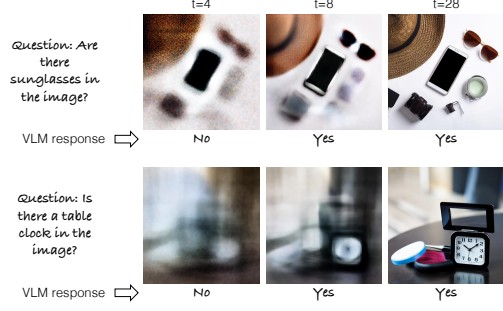

Figure 5: **Performance for each sub edit-type.** Training with only DMD loss fails to achieve certain tasks like removal and style changes. In addition, using binary cross-entropy loss and VLM identity-based questions helps improve the overall performance.

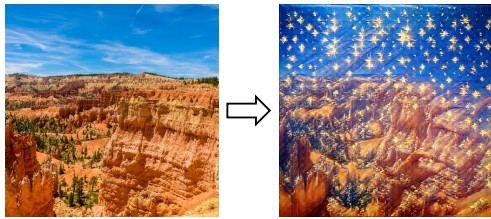
Replace the blue sky with a starry night sky

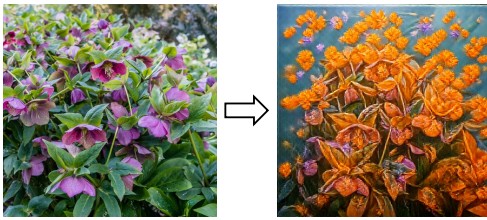
Replace the purple flowers with orange flowers

Figure 6: **Training with only VLM-editing loss** leads to lower fidelity samples with the model only maximizing the edit success probability. Current general-purpose VLMs are often not good at subjective tasks like evaluating image fidelity, highlighting the requirement of distribution matching loss in our framework.

Figure 7: **Unreliable VLM response on intermediate outputs of a multi-step diffusion model.** Here we show a 28-step diffusion process, denoising predictions from early steps (e.g., $t = 4$), which correspond to high noise levels, are blurry and semantically ambiguous. This can lead to unreliable responses from the VLM, as shown here. Therefore, we adopt a few-step diffusion model that always generates sharp images.

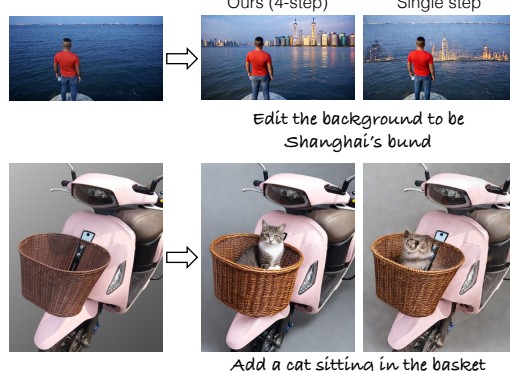
Edit the background to be Shanghai's bund

Add a cat sitting in the basket

Figure 8: **Our (4-step) vs single-step editing model.** We compare our final 4-step model with a single-step model, both trained via our approach. Editing an image in a single step is still challenging and leads to lower-fidelity outputs.

*Color change*, since DMD matches the text-conditioned score between the fine-tuned and pre-trained models, thus improving overall text alignment. However, it fails on tasks like *Removal*, underscoring the importance of VLM-based editing loss and its generalizability across diverse editing instructions. In addition, VLM-based loss also helps in maintaining better consistency between input and edited images (first row of Figure 4 in the main paper). However, when training with only the VLM-based editing loss, there's a gradual degradation in image quality, as Figure 6 shows, highlighting the complementary role of distribution matching losses such as DMD.

**Sampling steps.** For our method, we chose to train a few-step image-editing model instead of a multi-step diffusion model, as multi-step diffusion has a noisy or blurry estimate of the final output in the early stages of diffusion. This can make it difficult to get a reliable response from the VLM, as shown in Figure 7. On the other hand, predicting an edited image in one step is still challenging, as mentioned in the main paper, and shown here in Figure 8. Thus few-step provides a nice balance between the two extremes of single and multi-step sampling for our purposes.

**Sensitivity to prompt wording in VLM-based editing loss.** It has been observed that VLMs can be sensitive to the specific prompt wording (Zhou et al., 2022). While we did not need extensive prompt engineering, we find that removal edit-type benefits from explicitly asking about an object's presence in the *Edit-verification* question template, as mentioned in Section 4.2. To reiterate, the two templates are: (1) "Answer with a Yes or No if the image has {object name}" and (2) "The objective is to evaluate if the editing instruction has been executed in the second image. Editing instruction: {edit instruction}. Answer with a Yes or No.". Here, we show a qualitative comparison of two models trained for the removal edit-task while varying the *Edit-verification* question to the above two templates. As Figure 9 shows, the model trained using more specific prompt wording has better performance. Nonetheless, we expect such prompt-level sensitivity to diminish as VLMs continue to improve in robustness and reliability.

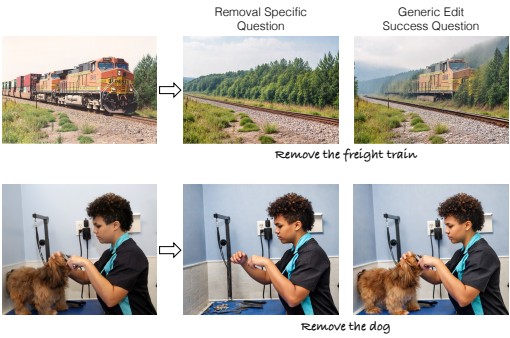

Figure 9: **Sensitivity to prompt wording.** We compare models trained for removal edit-type using two different template questions. A more specific question, such as, "Answer with a Yes or No if the image has {object name}", leads to better performance compared to a more general question, such as, "The objective is to evaluate if the editing instruction has been executed in the second image. Editing instruction: {edit instruction}. Answer with a Yes or No.".

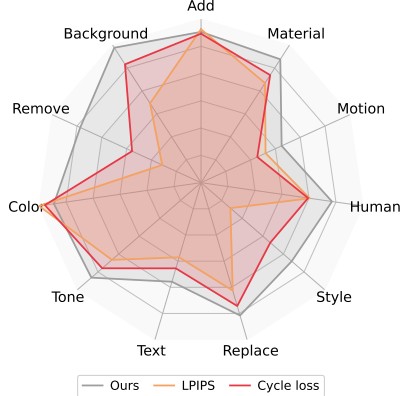

Figure 10: **Performance for each sub edit-type for LPIPS and cycle-loss on GEdit-Bench.** Additional regularization losses like LPIPS between input and edited image or cycle loss help increase the alignment between input and output edited image as shown in Figure 11, but lower the overall editing success.

## C    LIMITATION

**Alignment between input and edited image.** A limitation of our framework is the lack of pixel-wise supervision to preserve regions that should remain unchanged given an edit instruction. Consequently, edited images can deviate from the input image in details or spatial alignment, as shown in Figure 11, 1st column. While our VLM-based editing loss includes a question to check consistency between the input and edited images (Identity-preservation question in Section 4.2), it does not enforce pixel-level alignment, since current VLMs struggle to detect subtle changes and minor details.

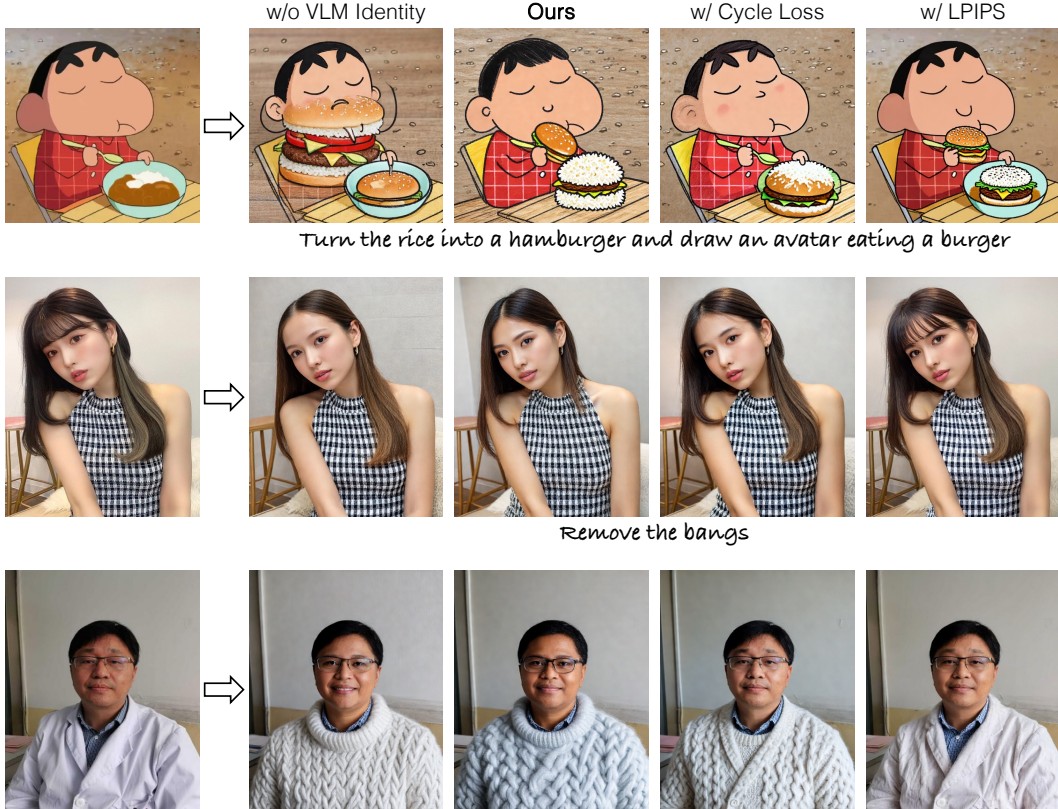

Figure 11: **Limitation.** We use an Identity-preservation question, Section 4.2, to maintain the consistency between input and edited image (1st vs. 2nd column). However, given the lack of pixel-level supervision, it can still struggle in many scenarios to maintain exact identity. Adding cycle loss between input and cycle-edited image helps mitigate this issue to an extent (3rd column), though with a drop in performance (Overall VIEscore of 5.10 vs 6.10 for Ours). We also experimented with LPIPS (Zhang et al., 2018) loss between the input and edited image, which helps (1st row) but at the cost of completely failing on tasks like removal (2nd row).

Here, we experiment with two regularization losses that can improve the alignment between input and edited images in regions that should not be edited: (1) LPIPS (Zhang et al., 2018) loss between input and edited image, and (2) Cycle loss (Zhu et al., 2017) between input and cycle-edited image.

In the cycle loss, for an edited image (e.g., remove the flower pot), we apply a reverse edit to it (e.g., add a flower pot) and encourage this reversal to reconstruct the original input image. We compute the reconstruction loss between the input and reverse-edited image in the SigLIP (Zhai et al., 2023) feature space. We select SigLIP's global embedding over fine-grained feature spaces like LPIPS to provide flexibility when reverse instructions are ambiguous (e.g., restoring the exact original object). For training with cycle loss, we add it to our training objective after 4K iterations. The reverse edit instruction for each (image, instruction) pair is generated using Qwen-32B VLM.

While LPIPS loss improves consistency, it degrades editing quality, particularly for edit-types like *Removal*, as shown in Figure 11. Cycle loss works better in comparison but still has lower edit success than ours, shown in Figure 10. The overall VIEScore on GEdit-Bench for LPIPS and cycle loss is 4.52 and 5.21 respectively, compared to 6.10 for Ours. However, we anticipate that as VLMs improve, with the capability of nuanced, pixel-level assessment, it will also help improve the input-edit alignment in our method without requiring additional regularization.

**VLM supervision for difficult editing tasks.** While NP-Edit can successfully edit images for various challenging edit-types like actions (2nd last row in Figure 2) and shape changes (Figure 12 (a)), our method is fundamentally upper bounded by the VLM's ability to understand and reason about image editing. Consequently, it currently fails at tasks requiring complex spatial reasoning (e.g., moving objects), as VLMs struggle to provide meaningful feedback in these scenarios. We show sample

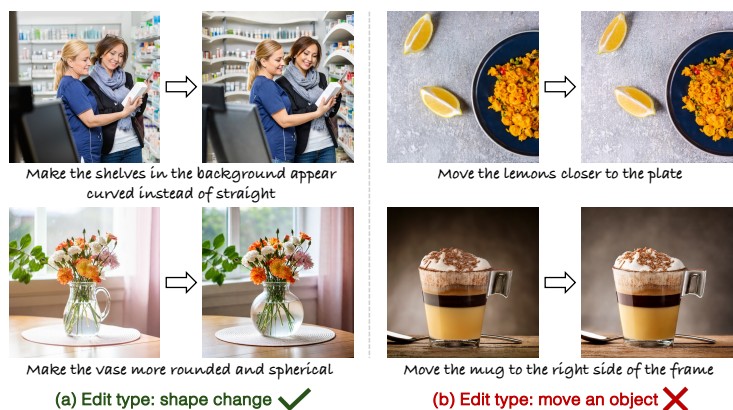

Figure 12: **Challenging and difficult editing tasks.** Our method can perform various challenging edits as well, like shape change as shown in (a). But we fail at other difficult tasks requiring spatial reasoning, such as moving objects, where VLM-based gradient feedback is less reliable.

failure case examples in Figure 12 (b). As unified generative models and Vision–Language Models advance, the instruction-following capabilities learned through our NP-Edit training framework are expected to improve correspondingly.

## D  DATASET CONSTRUCTION DETAILS

Each tuple in our dataset $\mathcal{X} = \{(\mathbf{y}_i, \mathbf{c}_i, \mathbf{c}_i^{\mathbf{y}}, \mathbf{c}_i^{\mathbf{x}})\}_{i=1}^{N}$ consists of a real reference-image, $\mathbf{y}$, corresponding edit instruction, $\mathbf{c}$, and text prompt corresponding to the reference and edited image, $\mathbf{c}^{\mathbf{y}}$ and $\mathbf{c}^{\mathbf{x}}$, respectively. We use a text-image dataset corpus to select reference images. Given a reference image, we prompt Qwen-2.5-32B VLM to suggest different possible editing instructions. The system and user prompt for it are as follows:

> **Role**: system, **Content**: You are a helpful assistant and an expert in image editing.
> **Role**: user, **Content**: Task: As a researcher in image editing, your task is to generate simple editing instructions based on the given image.
>
> The edit types you can use include: 1) local color change, 2) local texture, 3) adjust (shape change), 4) add, 5) remove, 6) replace, 7) bg, 8) style, 9) action, and 10) text manipulation
>
> **Important**: Ensure that you create a balanced distribution of these edit types when generating the instructions. Each example should utilize a different edit type, and the edit types should be evenly distributed across all examples.
>
> When using the "add" edit type, DO NOT USE vague placements like 'near', 'under', or 'beside', instead, specify the exact location where the object should be placed. For example, instead of "add a castle near the trees" use "add a castle in the clearing between the trees".
>
> Ensure that each instruction is straightforward and points to a single, clear edit change. Avoid complex or multi-step instructions.
>
> **Avoid Redundancy**: Make sure to introduce diversity in the edit instructions.
>
> Given the input image, could you generate simple edit instructions for different possible edit types by following the "format" of examples below and based on what you have seen in the image?
>
> Here are some examples showing the use of various edit types:
> Good example 1: {color change example}
> Good example 2: {texture change example}

Good example 3: {adjust shape example}
Good example 4: {add example}
Good example 5: {remove example}
Good example 6: {replace example}
Good example 7: {bg example}
Good example 8: {style example }
Good example 9: {action example}
Good example 10: {text manipulation example}

Bad Examples: the edit instructions are hard/impossible to perform well, or mention vague terms that make the editing model struggle to perform well, and you should not follow.
Bad example 1:
- Instruction: make this dog look like it's ready for a formal evening out?
- Type: add
- Reasoning: This instruction is bad because it does not mention the exact changes that are needed to make the dog look like it's ready for a formal evening out.

Bad example 2:
- Instruction: remove the balloon [given an image of only balloons on a white background]
- Type: remove
- Reasoning: This instruction is bad as it removes the only object in the image.

**Important Considerations**:
1. Avoid repetition of specific phrases: Do not reuse examples or themes from the above examples. Create entirely new and diverse themes and scenarios.
2. Logical Flow: Ensure that each instruction is logical and makes sense given the image.
3. Specificity in Insertions: When adding objects, use precise placement (e.g., "in the sky" or "on the lake"). Avoid vague terms like "next to", "around", or "near".

4. Balanced use of edit types: Use a variety of edit types such as [insertion], [replace], [local texture], [shape change], [style], [remove], [local color change], and [bg]. Ensure an even distribution of these edit types across your examples.
5. Diverse scenarios: Introduce variety in the scenarios, such as futuristic, historical, magical, surreal, or natural settings. Avoid overusing common tropes.
6. DO NOT suggest instructions that change a very small/minute part of the image.

Could you now generate 4 examples of **new, creative, and contextually relevant** edit instructions by following the format above? Avoid using the specific phrases, themes, or scenarios from the examples provided above.
**Each example must use a different edit type** from the ones listed above. Also, make sure to use each edit type equally across all generated examples.
Finally, you should make the edit instructions as simple as possible so that the downstream editing model is able to work well.

In the above user prompt, for the good examples, we randomly select an edit instruction for each editing type out of a fixed set of manually defined edit instructions. Given edit instructions for each image, we again prompt the VLM to check the validity of the edit instruction and, if valid, to suggest a possible caption for the edited image. The system and user prompt for this is:

**Role**: system, **Content**: You are a helpful assistant and an expert in image editing.
**Role**: user, **Content**: Task: As a researcher in image editing, given the input image, edit type, and the edit instruction, your task is to check if a given edit instruction is valid and can be applied to the image. If it is valid, generate a descriptive caption for what the image would look like after applying the edit instruction. If it is not valid, return "invalid" and explain why it is not valid, and output "NA" for the edited image caption.

An edit instruction is invalid if it:
1. mentions to modify/remove/replace an object that is NOT PRESENT in the image.
2. is TOO HARD to make editing model to understand and perform well, e.g., "remove any visible accessories."
3. DOES NOT change the image in any meaningful way, e.g., given the image of a forest, "change the background to a dense forest."

For the "remove" edit type:
- DO NOT mention the object that is removed during the edit in the edited image caption. For example, given an image of a cat in a living room on a sofa with the edit type "remove" and edit instruction: "remove the cat"
Bad Example: A cat is removed from the sofa in a living room.
Good Example: A living room with a sofa.

Given the edit instruction and the original caption:
Edit type: {edit type}
Edit instruction: {simple edit instruction}

Output format:
Validity: ...
Reasoning: ...
Edited image Caption: ...

Please provide a concise but complete caption describing the edited image. Focus on the changes that would be made according to the edit instruction.

Here are some more examples:
Example 1:
- Edit type: bg
- Edit instruction: change the background to a sunset view
- Validity: valid
- Reasoning: The edit instruction is valid because it adjusts the current blue sky to a sunset view, which is a meaningful change.
- Edited image caption: A park with a sunset view. People are walking around in the park.

Example 2:
- Edit type: remove
- Edit instruction: remove the wine glass
- Validity: invalid
- Reasoning: The edit instruction is invalid because it mentions removing a wine glass that is not present in the image.
- Edited image caption: NA

**Important Considerations**:
1. DO NOT use instruction words like replaced, added, removed, modified, etc. in the caption.
2. Keep the caption general to explain any possible images resulting from the edit instruction.

Only output the validity, reasoning, and edited image caption. Do not include any other text or explanations.

After filtering the list of generated editing instructions using the above procedure, our final dataset consists of approximately 3M unique reference images with corresponding editing instructions spanning the 10 edit sub-types. Within the constraints of our available computational resources, this represents the largest dataset we were able to construct.

For the customization task, we first instruct the VLM, to identify if the image has a prominent object in the center. We provide an in-context sample image as well to the model. The exact system and user prompt for this is:

**Role**: system, **Content**: You are a helpful assistant and an expert in image personalization/customization.

**Role**: user, **Content**: Task: You are assisting in a research project on image personalization. Your goal is to evaluate whether the SECOND image contains a **single, uniquely identifiable object** prominently positioned near the center of the frame.

- The FIRST image (image˙path1) is an example of a valid case.
- The specific object category in the second image can be different — focus only on **object uniqueness** and **image composition**.

Good examples include object categories that can be personalized, have unique texture, and are not general objects:
- Backpack, purse, toy, cat, dog, cup, bowl, water bottle, wearables, plushies, bike, car, clocks, etc.

Bad examples include object categories that are general objects, and different instances of the category can not be distinguished:
- Tree, building, door, flowers, food, vegetables, fruits, natural scenes, roads, etc.

**Important Considerations:**
1. The object should be clearly recognizable and **visually distinct** from the background.
2. The object should be **near the center** of the image.
3. The **entire object** should be visible — it should NOT be a tight or zoomed-in crop.
4. The background can be natural but should not be overly cluttered or visually distracting.
5. The image should feature a **single primary object**, not multiple equally prominent objects.

Could you now judge the SECOND image and only provide the output, reasoning, and object name, in the following format:
Output: True/False
Reasoning: Brief explanation
Object Name: The name of the object (e.g., "backpack", "cat", "toy").

If the VLM response predicts a valid image, we then query it again to suggest a new background context for the object category as follows:

**Role**: system, **Content**: You are a helpful assistant and an expert in image personalization/customization.

**Role**: user, **Content**: Given an image of an object category, you have to suggest three DIVERSE background captions for the object. Provide a detailed description of the background scene. Only suggest plausible backgrounds. DO NOT add the object name in the caption. DO NOT use emotional words in the caption. Be concise and factual but not too short. DO NOT mention the object name in the output captions. If the object is not a thing, but a scene, then output None.

Example background captions for "White plastic bottle" are:
1. near the edge of a marbled kitchen counter, surrounded by a cutting board with chopped vegetables, a salt shaker, and a stainless steel sink in the background.
2. rests on a tiled bathroom shelf, accompanied by a toothbrush holder, a mirror with foggy edges, and a shower curtain partially drawn open.

Example background captions for "a blue truck" are:
1. parked beside a graffiti-covered brick wall under a cloudy sky, with city skyscrapers rising in the background.
2. resting in a grassy field surrounded by wildflowers, with distant mountains and a golden sunset in the background.

> Object: {object category name}
> Output:
> 1.
> 2.
> 3.

# E  TRAINING IMPLEMENTATION DETAILS

## E.1  LOCAL-IMAGE EDITING

**Training hyperparameters.**  We train on a batch-size of 32 using Adam (Adam et al., 2014) optimizer with a learning rate of $2 \times 10^{-6}$, $\beta_1$ as 0, and $\beta_2$ as 0.9, on 32 A100-80GB GPUs. The few-step sampling time schedule, $[t_4, t_3, t_2, t_1]$ is set to $[1.0, 0.90, 0.70, 0.47]$, shifted towards more noisy region (Esser et al., 2024). We train for a total of 10K iterations with $N_{\text{aux}} = 10$, i.e., auxiliary network, $A_\phi$, being updated 10 times for every generator, $G_\theta$, update, following similar strategy adopted in DMD2 (Yin et al., 2024a). We train with the identity loss (Section 4.3) for 250 iterations. For faster convergence, the first 4K training iterations are trained with a single step prediction ($t = 1$ in Line 3 of Algorithm 1), and then we start the 2-step unrolling of the diffusion trajectory. The final loss is a weighted combination of VLM-based editing loss and distribution matching loss with $\lambda_{\text{VLM}} = 0.01$ and $\lambda_{\text{DMD}} = 0.5$. During training, we also add a "do nothing" editing task with $\mathcal{L}_2$ loss between the input and edited image as regularization with a $1\%$ probability. This helps the model learn to maintain consistency between input and edited images. During training, we sample the editing instruction corresponding to each subtype uniformly, except *removal*, which is sampled with $25\%$ probability. This is because, empirically, we observe that *removal* is more difficult than other edit-types like *Color change*.

**Template questions for VLM-based editing loss.**  As mentioned in the main paper, we evaluate VLM-based loss on two questions per edit type. Specifically for any edit type except removal, we use the following template:

> **Role**: user, **Content**: You are a professional digital artist and an expert image editor. You will be provided with two images.
>
> The first being the original real image, and the second being an edited version of the first. The objective is to evaluate if the editing instruction has been executed in the second image.
>
> Editing instruction: {edit instruction}
>
> Answer with a Yes or No.
> Note that sometimes the two images might look identical due to the failure of image editing. Answer No in that case.

> **Role**: user, **Content**: You are a professional digital artist and an expert image editor. You will be provided with two images.
>
> Answer with a Yes or No if the second image is exactly the same as the first image. IGNORE the changes in the second image because of the edit: {edit instruction}. Everything else should be the same.

For the removal edit-type, we change the first question to explicitly ask about the presence of the target object to be removed, with the ground truth answer in this case being *No*. We find it to be more effective than a generic template.

> **Role**: user, **Content**: You are a professional digital artist and an expert image captioner. You will be provided with an image.
> Answer with a Yes or No if the image has {object name}.

### E.2 Free-form editing (Customization)

**Training hyperparameters.** We reduce the warmup iterations for which we train with the identity loss to $100$ in this case, since customization often requires more drastic changes in the output image compared to the input reference image. Further, we increase $\lambda_{\mathrm{DMD}} = 2$ instead of $0.5$ as in the case of local image-editing. The rest of the hyperparameters remain the same. Both during training and inference, the input text prompt to the few-step generator, $G_\theta$, is in the following template: `Generate the main object shown in the first image in a different setting and pose: { background scene description}`. We train the 4-step model for $10K$ iterations. For the 8-step model, we fine-tune for $5K$ additional training steps starting from the 4-step model.

**Template questions for VLM-based editing loss.** Here, we modify the questions to instead evaluate if the background context and pose are different in the generated image, i.e., editing success, and if the object identity is similar, i.e., image alignment and consistency between the input reference and edited image. The exact questions are as follows:

> **Role**: user, **Content**: You are a professional digital artist and an expert in image editing. You will be provided with two images.
>
> Answer with a Yes or No if the {object name} in the second image is in a different pose and location than in the first image. Note that sometimes the second image might not have the same object because of the failure of image editing. Answer No in that case.

> **Role**: user, **Content**: You are a professional digital artist and an expert in image editing. You will be provided with two images.
>
> Answer with a Yes or No if the {object name} in the second image is the exact same identity, with similar color, shape, and texture as in the first image. Note that sometimes the second image might not have the same object because of the failure of image editing. Answer No in that case.

## F Other Baseline Details

**Flow-GRPO (Liu et al., 2025a).** We follow the open-source implementation of Flow-GRPO and train with the same computational budget as our method, i.e., across 4 A100 GPU nodes (32 GPUs) and $2.5$ days of training. The final model is fine-tuned from a pre-trained image-editing model for $5K$ iterations. During training, we collect 16 images per-prompt with 12 denoising steps (28 during inference) for computing the mean and standard deviation in GRPO (Shao et al., 2024). Following their official implementation, we train with LoRA (Hu et al., 2022) of rank 32, $\alpha = 64$, learning rate $1 \times 10^{-4}$, and use the VLM to score edits on a scale of 0 to 9, which is normalized between 0-1 to get the final reward. The exact prompt used to query the VLM is derived from VIEScore (Ku et al., 2024) and is shown below.

> **Role**: system, **Content**: You are a helpful assistant and an expert in image editing.
> **Role**: user, **Content**: You are a professional digital artist. You will have to evaluate the effectiveness of AI-generated edited image(s) based on given rules.
>
> You will have to give your output in this way (Keep your reasoning VERY CONCISE and SHORT):

> score : ...,
> reasoning : ...
>
> RULES:
> Two images will be provided: The first being the original real image and the second being an edited version of the first.
> The objective is to evaluate how successfully the editing instruction has been executed in the second image.
>
> Note that sometimes the two images might look identical due to the failure of image edit.
>
> From scale 0 to 9:
> A score from 0 to 9 will be given based on the success of the editing. (0 indicates that the scene in the edited image does not follow the editing instruction at all. 9 indicates that the scene in the edited image follows the editing instruction text perfectly.)
>
> Editing instruction: {edit instruction}

**Supervised Fine-Tuning.** We train with the standard velocity prediction flow-objective for 30K iterations with a batch-size of 32 and learning rate $2 \times 10^{-6}$ with a linear warmup of 2K iterations. To enable classifier-free guidance, we drop the image and text conditions $10\%$ of the time.

**Sampling parameters for local image-editing baselines.** We follow the open-source implementation to sample images from all the baseline models for the benchmark evaluations. The turbo-edit (Deutch et al., 2024) baseline requires a caption corresponding to the edited image as well, and we use Qwen-2.5-32B-VLM to generate these captions for GEdit-Bench images.

**Sampling parameters for customization baselines.** We follow the open-source implementation to sample images from all the baseline models for the benchmark evaluations. In the case of DSD (Cai et al., 2025), it employs Gemini-1.5 to convert the input user-prompt into a detailed prompt. However, we skipped this step for a fair evaluation with other methods, which do not use any prompt rewriting tools. In the case of SynCD (Kumari et al., 2025), though it supports multiple reference images as input, we evaluated it with a single reference image, to keep the setup similar to other baseline methods and Ours. For sampling images with OminiControl (Tan et al., 2025) and DSD (Cai et al., 2025), we follow their recommended prompt setting and replace the category name with "this item".

## G  USE OF LLM IN PAPER WRITING.

We primarily use ChatGPT as a grammatical aid and for polishing paragraphs, instructing it to "refine the text without adding or omitting any relevant information."

## H  REPRODUCILITY OF RESULTS

We provide full implementation details of our method, including dataset construction, training algorithm, and hyperparameters in Section 4 and Appendix D, E. A pseudo-code description is also included in Algorithm 1. Together, these are intended to facilitate the reproducibility of all results reported in our paper.

## I  ETHICS STATEMENT

Our work introduces a training framework for fine-tuning text-to-image models into a *few-step* image-editing model without using paired supervision. By enabling few-step sampling, our method improves inference efficiency and reduces computational cost. Nonetheless, the broader risks of generative models, such as creating deepfakes and misleading content, also apply to our approach. Possible ways to mitigate this are watermarking generative content Fernandez et al. (2023) and reliable detection of generated images Wang et al. (2020); Corvi et al. (2023); Cazenavette et al. (2024)

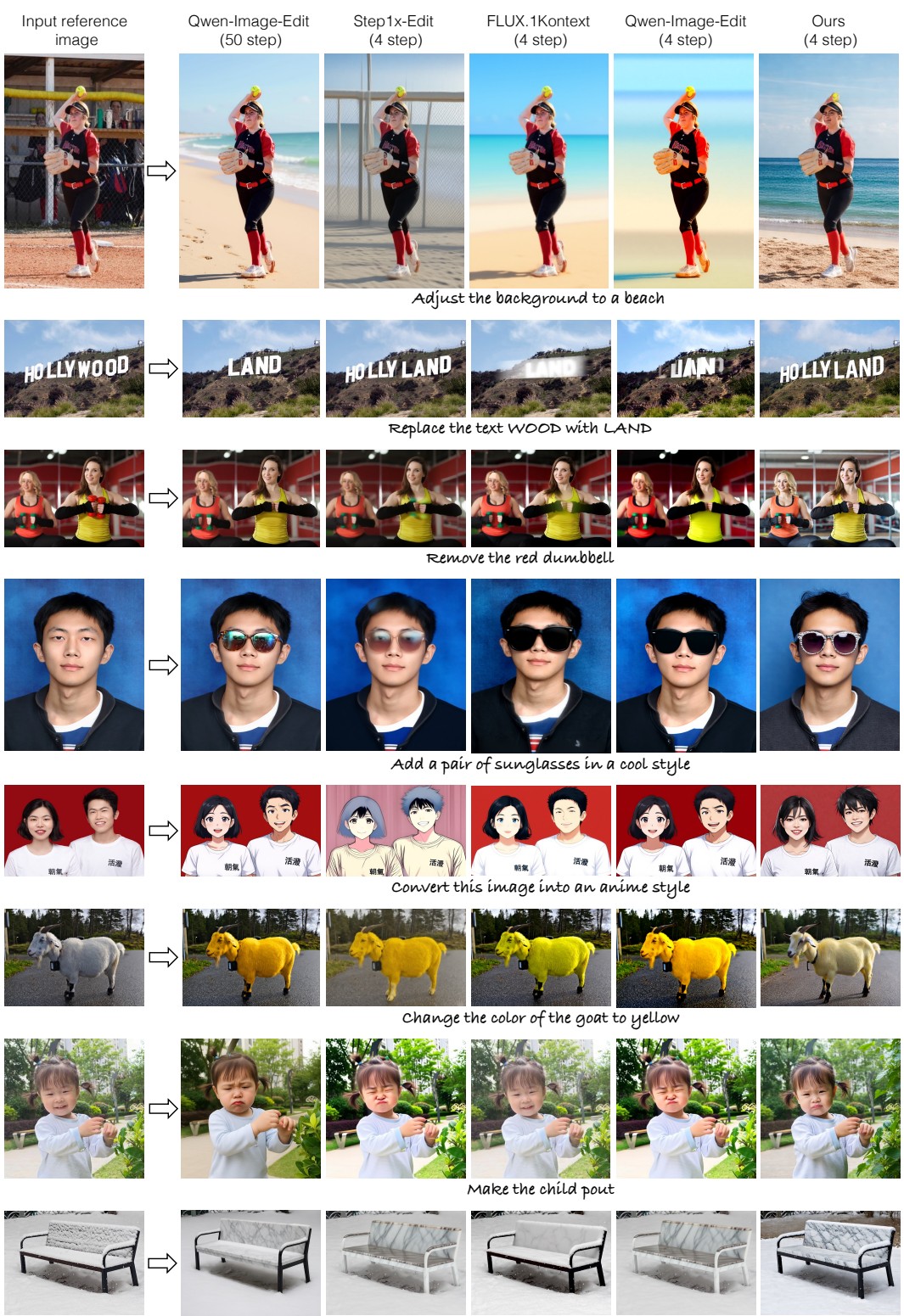

Figure 13: **Qualitative comparison on GEdit-Bench.** We show results of our and baseline image-editing methods under the few-step sampling setting. For comparison, we also show the results of the best method with multi-step sampling, as measured by the quantitative metrics (Table 1), in the [1]st column. Our method performs on par or better than baseline methods across different edit types in the few-step setting.

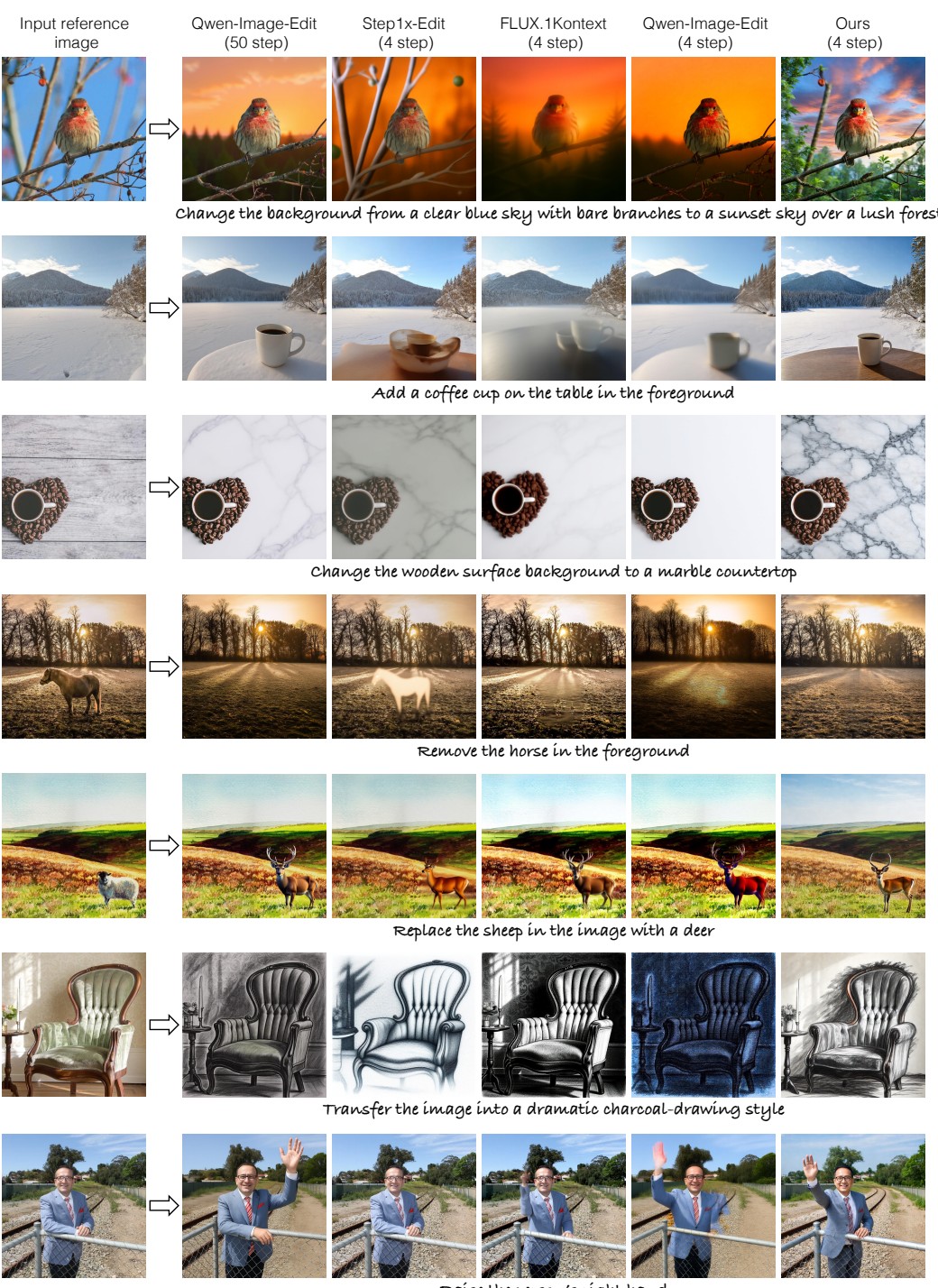

Figure 14: **Qualitative comparison on ImgEdit-Bench.** We show results of our and baseline image-editing methods under the few-step sampling setting. For comparison, we also show the results of the best method with multi-step sampling, as measured by the quantitative metrics (Table 7), in the 1st column. Our method performs on par or better than baseline methods across different edit types in the few-step setting.

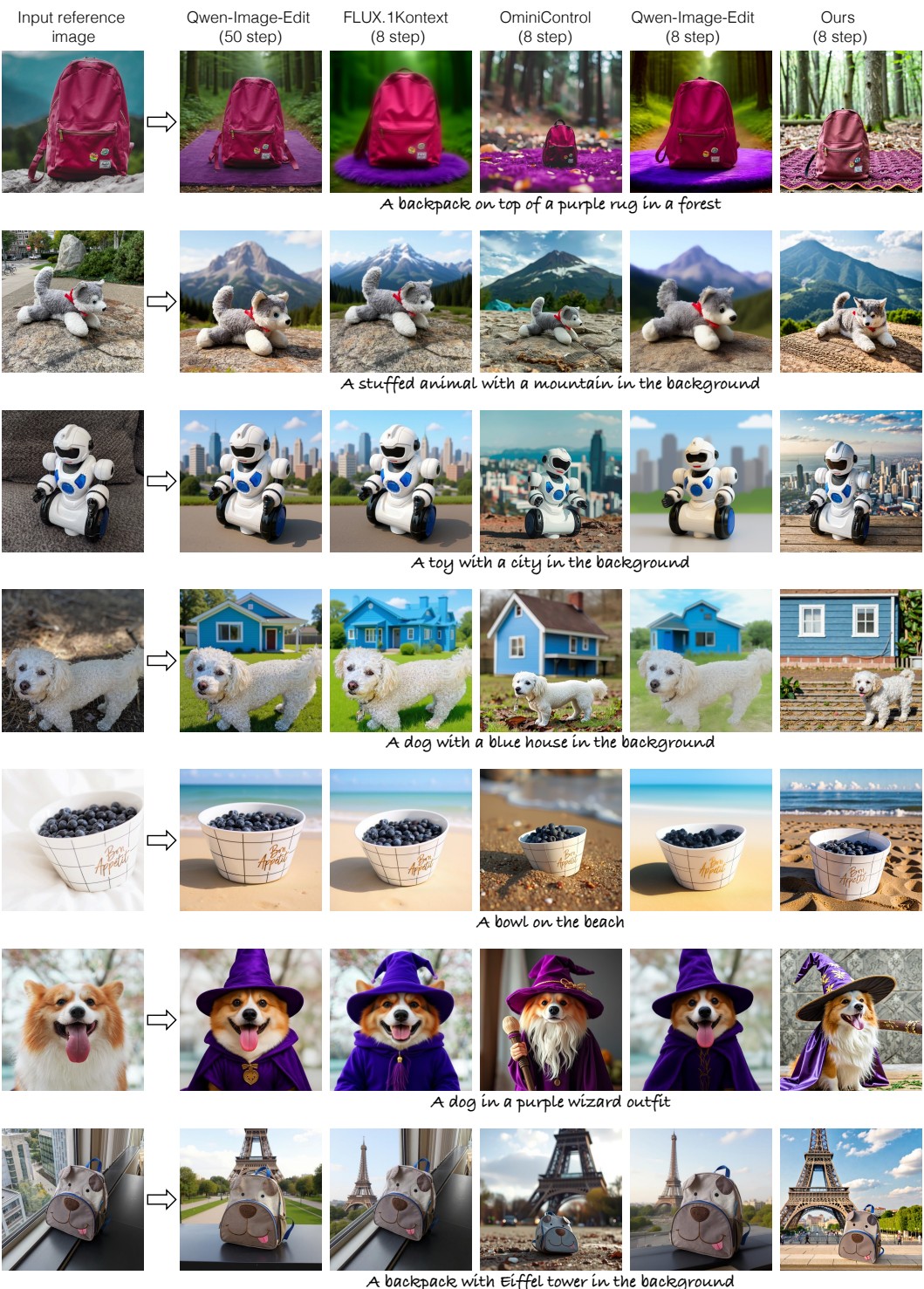

Figure 15: **Qualitative comparison on DreamBooth.** We show results of our and baseline methods under the few-step sampling setting. For comparison, we also show the results of the best method with multi-step sampling, as measured by the quantitative metrics in the first column. Our method performs comparably with baseline methods on identity alignment while having better image fidelity across different concepts in the few-step setting.

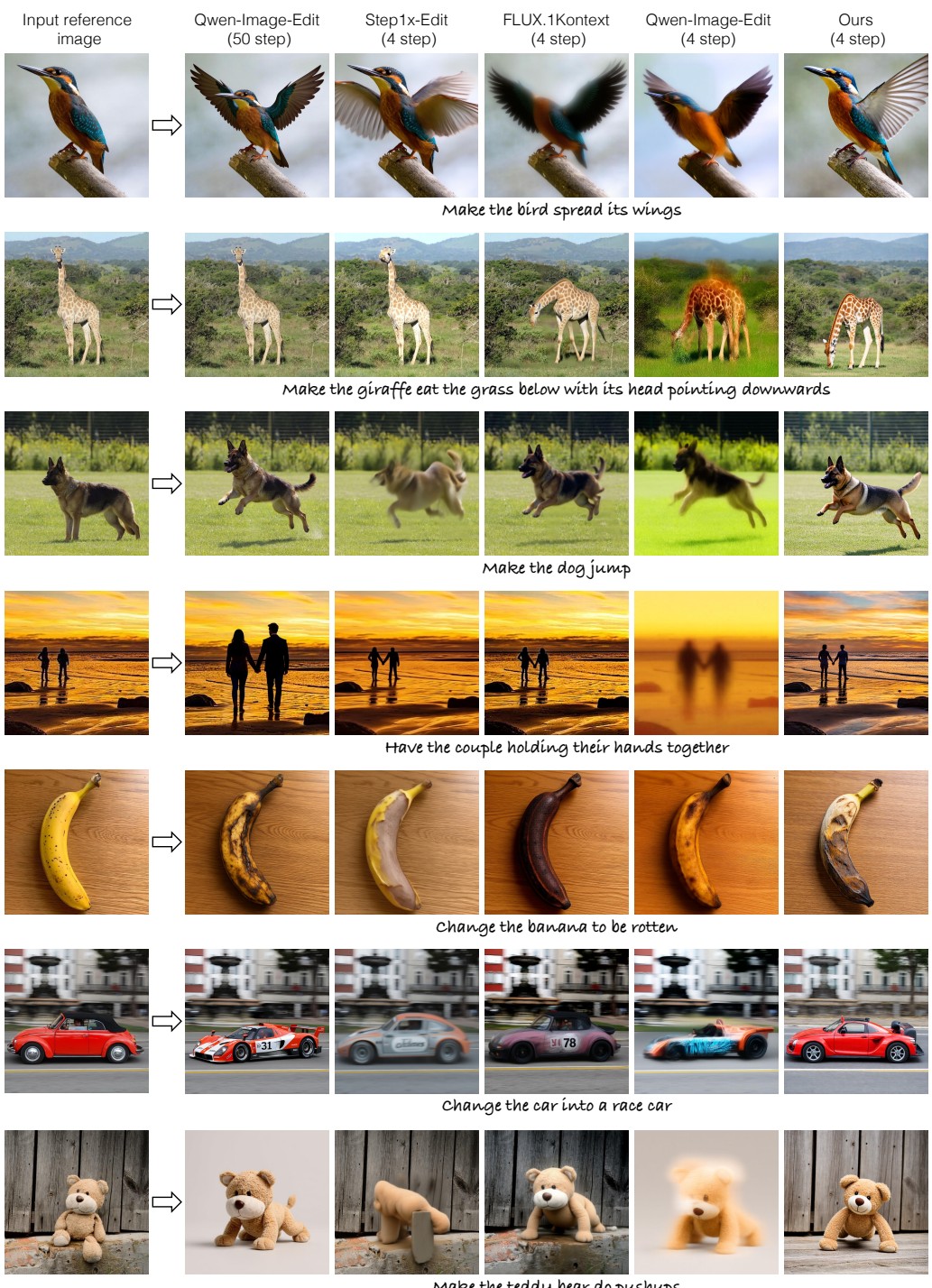

Figure 16: **Qualitative comparison on TED-Bench.** We show results of our and baseline image-editing methods under the few-step sampling setting. For comparison, we also show the results of the best method with multi-step sampling, as measured by the quantitative metrics (Table 8), in the 1st column. Our method performs on par or better than baseline methods across different edit types in the few-step setting.

