# OpenReview forum: "Learning an Image Editing Model without Image Editing Pairs"
_ICLR.cc/2026/Conference — ICLR 2026 Poster_

### Official Review · Reviewer_JFwR · 2025-10-20

**Soundness:** 3
**Presentation:** 3
**Contribution:** 3
**Rating:** 6
**Confidence:** 4

**Summary:**

This paper proposes NP-Edit, a training paradigm for instruction-following image editing that removes the need for paired before/after supervision. The core idea is to unroll a few-step diffusion generator during training and optimize it end-to-end with differentiable feedback from a vision–language model (VLM) that answers templated Yes/No questions about (i) whether an edit was executed and (ii) whether identity/context was preserved; a distribution-matching distillation (DMD) loss to a text-to-image teacher constrains realism. The method trains a lightweight 2B-parameter few-step editor (4 steps by default), with results reported on GEdit-Bench and DreamBooth customization showing competitive performance with larger supervised baselines under few-step sampling, plus ablations showing the complementary roles of VLM feedback and DMD.

**Strengths:**

* **Originality.** The paper is, to my knowledge, the first to train a general instruction-following image editor using **gradient** feedback from a VLM (rather than scalar RL rewards or hand-crafted objectives), coupled with DMD to keep edits on the teacher manifold; this is a non-trivial and timely contribution in the post-training space.
* **Quality.** The technical formulation is clear: a two-step unroll to produce cleaner intermediate states for reliable VLM judgments; a binary-logit loss over Yes/No tokens; and a KL-motivated DMD term implemented with an auxiliary velocity predictor. Ablations convincingly show that removing either VLM loss or DMD substantially degrades performance.
* **Clarity.** The paper supplies an end-to-end algorithm box, precise templates for VLM prompts/questions, and concrete training hyperparameters (optimizer, learning rate schedule, warm-up, weighting, sampling times), all of which aid reproducibility.
* **Significance.** Competitive results with a 2B few-step editor against much larger systems under few-step sampling, and solid customization performance, suggest a practical path toward data-efficient editing without paired supervision. The explicit focus on few steps is also valuable for latency-sensitive applications.

**Weaknesses:**

* **Evaluation dependence on automated judges.** Most quantitative evaluation relies on VIEScore (GPT-4o-based) and related automated metrics; no human perceptual study is presented. This raises concerns about metric bias and circularity (a VLM both trains and evaluates), especially for nuanced fidelity judgments. Adding a user study or double-blind human pairwise comparisons would strengthen claims.
* **VLM supervision reliability and calibration.** The paper shows VLM judgments are unreliable on noisy/blurry intermediates—motivating few-step training—but broader issues remain: sensitivity to prompt wording and model-specific biases. More analysis of calibration or robustness (e.g., multiple VLM judges, temperature/decoding settings) would be helpful.
* **Resource/engineering overhead.** Keeping a VLM in GPU memory is noted as a limitation; concrete throughput/memory numbers for typical hardware (and at training vs inference) are not reported, which makes it harder to assess deployability.

**Questions:**

* **Judge diversity.** Beyond LLaVA-OneVision-7B, what happens if you ensemble multiple VLMs (or mix families) as teachers during training? The scaling table shows benefits from stronger VLMs; an ensemble could mitigate single-judge biases—did you attempt this?
* **Human evaluation.** Can you provide a small-scale user study (e.g., 100 instructions × 3 systems, pairwise preferences for edit correctness and fidelity) to corroborate VIEScore-based gains and address potential metric bias?
* **Failure modes by edit type.** The ablations suggest that Removal and certain style edits are harder. What specific misbehaviors occur (e.g., partial removal, texture spillover), and can targeted question templates or auxiliary losses mitigate them?

---

> ### Author Response · Authors · 2025-11-24
> **Response to reviewer JFwR**
>
> We thank the reviewer for their detailed comments and for finding our method non-trivial with a clearly written formulation. Below, we address all the individual questions:
>
> ### **VIEScore metric's correlation with Human evaluation (W1)**
> VIEScore has been shown to correlate with human judgment across various image generation tasks. To verify this in our setting, and following your suggestion, we conducted a pairwise human preference study on GEdit-Bench under the few-step sampling setting (~500 responses per comparison).
>
> The resulting preferences for editing quality align well with the VIEScore trends reported in Table 1: with our method outperforming Turbo-Edit and remaining competitive with other baselines, with a slight preference over FLUX.1-Kontext and Step-1X Edit, and close to random chance with Qwen-Image-Edit, despite being much smaller in parameter count and not utilizing any paired training data. We have also updated Appendix A with these results.
>
> | vs | TurboEdit | FLUX.1-Kontext | Step-1X Edit | Qwen-Image-Edit |
> |:---|:---:|---:|---:|---:|
> | Ours (Editing quality) | 63.19 % | 53.35 % | 56.46% | 48.44 % |
> | Ours (Image quality) | 73.92% | 79.91 % | 80.39 % | 71.98 % |
>
>
> ### **VLM supervision reliability and calibration (W2)**
> Though our method shows the viability of using gradient feedback from VLMs for general-purpose editing tasks, it is obviously limited by the current VLM’s capabilities. As suggested, we analyze some of these limitations and sensitivity analysis regarding VLM-feedback:
>
> **Effectiveness on difficult edit types**: While we can already perform various challenging edits such as object addition and shape change, as shown in Figure 2, 2nd last row, and new Figure 11 (a) in the Appendix. However, it fails at tasks requiring complex spatial reasoning (e.g., moving objects), as current VLMs struggle to provide meaningful feedback in such scenarios. We have added such sample failure cases in **Appendix Figure 11** (b).
>
>
> **Sensitivity to prompt wording**: While we did not need extensive prompt engineering, we observed that removal edit-type benefited from explicitly asking about an object's presence rather than the generic templates used for other edit-types (as mentioned in Lines 199–201). We have added a detailed analysis in Appendix B, where **Figure 9** provides a qualitative comparison between the two prompt templates. Nonetheless, we expect such prompt-level sensitivity to diminish as VLMs continue to improve in robustness and reliability.
>
>
> **Sensitivity to temperature**:  Since we run a single forward call to the VLM and use the predicted logits directly to compute our loss, the VLM’s autoregressive sampling temperature parameter is not used during our training.
>
>
> **Ensemble of VLMs**: We agree that this is a promising direction, but at present limited by GPU memory constraints to perform this experiment.
>
>
> ### **Computational resources and scalability (W3)**
> **Training**: Our model training requires ~1600 GPU hours (32xA100 with 80GB VRAM for ~50h) at 512x512 resolution images. This is comparable to open-source baselines such as OmniGEN (trained on 104×A800 GPUs). We have added this to our implementation details section in Appendix E.
> **Inferece**: Our inference is quite efficient because of the few-step sampling: 4-step edit at 512x512 resolution takes 0.41 seconds on a single A100 GPU.

---

> > ### Comment · Reviewer_JFwR · 2025-11-24
> > **Response to authors**
> >
> > Thank you for the detailed response and additional experiments. You have addressed nearly all of my concerns; the remaining point on more extensive VLM integration/ensembling is understandably left for future work given current GPU constraints. I have therefore raised my score to reflect my support for this submission, and I hope this line of research will further inspire the community.

---

> > > ### Author Response · Authors · 2025-11-24
> > > **Official Comment by Authors**
> > >
> > > Thanks for your comments and updated scores.

---

### Official Review · Reviewer_kKVu · 2025-10-29

**Soundness:** 3
**Presentation:** 3
**Contribution:** 3
**Rating:** 8
**Confidence:** 4

**Summary:**

This paper introduces NP-Edit, a new framework for training image editing diffusion models without any paired supervision. Instead of relying on input–output image pairs or synthetic datasets, NP-Edit leverages direct gradient feedback from Vision–Language Models (VLMs) to guide model optimization. The VLM evaluates whether an edit instruction is followed and whether unchanged regions are preserved, providing differentiable supervision. To maintain realism, the method also employs a Distribution Matching Distillation (DMD) loss to align generated images with the manifold of a pretrained diffusion model. The proposed framework achieves performance on par with supervised image editing diffusion models, despite requiring no paired data. The paper includes extensive experiments and ablation studies examining the influence of VLM backbone choice, dataset diversity, and loss formulation.

**Strengths:**

1. The paper addresses a well-recognized bottleneck in image editing, dependence on expensive, hard-to-scale paired datasets. The motivation is timely and relevant, given the increasing reliance on generative models in open-world applications.

2. The idea of replacing supervised pairs with gradient-based supervision from a VLM is conceptually elegant. This formulation could generalize beyond image editing to other multimodal generation tasks.

3. Competitive empirical performance under the few-step diffusion setting has been demonstrated.

4. The author also includes solid ablation studies and well-motivated analyses.

**Weaknesses:**

1. Since supervision is entirely derived from VLM feedback, the model’s performance is directly tied to the accuracy, bias, and robustness of the chosen VLM. Any systematic bias (e.g., cultural or aesthetic preferences) or failure to interpret nuanced edit instructions could propagate into the trained model.

2. Unrolling the few-step diffusion model during training and backpropagating through the VLM introduces potentially high computational and memory costs. The paper lacks a detailed analysis of training efficiency or scalability to high-resolution datasets.

**Questions:**

Overall, this submission makes a conceptual and practical contribution to the field of text-guided image editing by eliminating dependence on paired training data through the use of VLM-based differentiable supervision. The approach is both innovative and impactful, offering a scalable alternative to traditional supervised paradigms. The authors are suggested to address the above weaknesses to further strengthen the work.

---

> ### Author Response · Authors · 2025-11-24
> **Response to reviewer kKVu**
>
> We thank the reviewer for the encouraging comments and for acknowledging our work as timely and relevant. We address all the concerns and weaknesses in detail below:
>
> ### **Dependency on VLM’s strengths and limitations (W1)**
> **VLM bias**: We share the concern that the model may inherit underlying biases when relying on VLM feedback, and have added this as one of the limitations of our method in Section 5. However, this is part of a broader challenge: data-driven learning methods are often biased by the training dataset or by teacher models like VLMs in our case. To mitigate this to some extent, DMD plays an important moderating role by keeping generations close to the pretrained model’s manifold, reducing drift toward VLM-specific aesthetic or cultural biases. Moreover, as VLMs continue to improve with fewer systematic biases, our framework will naturally benefit from these improvements.
>
> **VLM's accuracy on nuanced and difficult edit instructions**: Our method can successfully perform various challenging edits, including non-rigid action and shape changes, Figure 2, 2nd last row, and new Figure 11 (a) in the Appendix. But it struggles with complex spatial reasoning (e.g., moving objects), as current VLMs struggle to provide meaningful feedback in such scenarios. We have added such sample failure cases in **Appendix Figure 11** (b).
>
> We also observe better performance with larger VLM backbones (Table 4). Thus, as unified generative models and Vision–Language Models advance, the instruction-following capabilities learned through our NP-Edit training framework are expected to improve correspondingly.
>
>
> ### **Computational resources and scalability (W2)**
> **Training compute**: Our model training requires ~1600 GPU hours (32xA100 with 80GB VRAM for ~50h) at **512x512 resolution** images. This is comparable to open-source baselines such as OmniGEN (trained on 104×A800 GPUs). We have also added this to our implementation details section in Appendix E.
>
> **Scalability to higher resolution**: Our framework can also be easily extended to higher resolution images, without a disproportionate increase in compute resources when compared to the standard diffusion model training. As VLM-feedback can be computed at a lower resolution, while the generative model operates at a higher target resolution.
>
> **Diffusion unrolling**: Regarding backpropagating through the diffusion sampling steps, we only require a maximum of two-step unroll, which helps keep the training procedure relatively efficient while providing the benefits of few-step inference.
> To reduce memory cost, we can also use stopgrad after the first step when training on intermediate timesteps with the two-step unroll. This prevents backpropagation through the unrolled diffusion trajectory, and only leads to a marginal drop in performance, with an Overall VIEScore of 5.96 compared to 6.1 for ours.

---

### Official Review · Reviewer_jdS1 · 2025-10-31

**Soundness:** 2
**Presentation:** 3
**Contribution:** 3
**Rating:** 4
**Confidence:** 3

**Summary:**

This paper aims at eliminating the image pairs which are requested by contemporary in-context DiT model, which are expensive to curate and scale. Given only a reference image and an instruction, the NPEdit model is supervised by the VLM's feedback signals. The authors further designed VLM-based editing loss and DMD loss. Rich experiments are conducted and NPEdit achieves state-of-the-art results under few-step setting.

**Strengths:**

1. The topic of eliminating the requirements of image pairs  is very important for In-context DiTs in image editing task. Because image-text pair are easy and cheap to collect and scale up. Instead, image pairs are expensive and rare.

2. The model proposes VLM-editing loss and DMD loss, which seems novel.

3. The results surprise me because according to my personal experience, training in-context DiTs with  image-text pairs usually leads to copy-paste problem. Yet the results on image customization task demonstrate that NPEdit could alter the spatial orientation and location quite well, which is interesting.

**Weaknesses:**

1. Inadequate experiments on non-rigid editing, which is refered to as 'action' in the paper. NPEdit only demonstrates one non-rigid editing, 'a person waves a hand', which is not persuasive. In a previous benchmark TEdBench[1], Imagic + Imagen[1]  and Forgedit + SD 1.4 [2] could conduct some hard non-rigid instructions on this benchmark, for example,  let the dog sit, jump, or  let a bird spread its wings, let a bird looking backward, close an open book etc.

2. Facial identity preservation does not seem good. VLMs are generally not good at measuring facial similarity, far worse than specific face recognition models like arcface. Using VLMs' judgements as supervision signals may not be accurate. The facial identities in the editing results of NPEdit further confirm this doubt.

3. The paper did not explain very clearly how to obtain the editing result fed to VLM during training. It seems like an inversion method. This step is very important in the training process and should be elaborated further.


Reference:

[1] Imagic: Text-Based Real Image Editing with Diffusion Models

[2] Forgedit: Text Guided Image Editing via Learning and Forgetting

**Questions:**

I  notice that there is  only one  non-rigid editing instruction ( which is referred as 'action') in the paper, " a person waves a hand". How about other non-rigid instructions? In TEDBench[1] editing benchmark,  Imagic[1] and Forgedit[2] could conduct various non-rigid editing, for example, let the dog sit/jump, or  let a bird spread its wings, let a bird look backward, close an open book etc.

I am willing to adjust my rating if  the authors could  address my concerns.

---

> ### Author Response · Authors · 2025-11-24
> **Response to reviewer jdS1**
>
> Thank you for your detailed feedback and for finding our method novel with interesting results. Below, we address the concerns regarding limited validation on non-rigid edit types and other questions:
>
> ### **More validation on non-rigid edit types (W1)**
> As suggested, we expanded our validation on more non-rigid/action edits, such as making a dog jump or a bird open its wings, using **TED-Bench**. We have included these results in the new **Figure 13 and Table 8 in the Appendix**. The quantitative results on TED-Bench show a similar trend to other benchmark evaluations: our method remains competitive with many multi-step baselines while maintaining its few-step efficiency.
>
> We emphasize that our method is a general-purpose framework for a variety of edit types, provided the edit type instructions are represented in the training data and the VLM can correctly assess the generated edited image.
>
> ### **Facial identity preservation (W2)**
> We agree that maintaining facial identity can be challenging, and we discussed this in our method's limitation (Lines 479-481) and Appendix C. We also demonstrated in **Appendix C, Figure 10** that adding pixel-supervision losses, such as LPIPS, between the input and edited output image resolves this to an extent, but harms edit types such as removal.
>
> An approach to address this can be to leverage specialized networks like ArcFace, which have been used to maintain facial identity in previous works, such as InstantID [1]. Using an ensemble of specialized judges like ArcFace with more general-purpose VLMs is a promising future direction.
>
> [1] Wang et al. InstantID: Zero-shot Identity-Preserving Generation in Seconds
>
>
> ### **Procedure for edited image fed to the VLM (W3)**
> To clarify, our training procedure does not perform inversion. Given the input condition image, y, and edit instruction, c, we follow Eq. 1 to obtain a clean edited image (x_0) via a two-step diffusion sampling starting from random noise. Since we are training a few-step model, sampling via two steps during training already results in a clean, non-blurry edited image, x_0. This is then fed to the VLM to assess editing success, and the loss is backpropagated all the way through the two unrolled diffusion sampling steps. We have also **updated Figure 1** to illustrate this process.

---

### Official Review · Reviewer_anZe · 2025-11-01

**Soundness:** 4
**Presentation:** 3
**Contribution:** 3
**Rating:** 6
**Confidence:** 4

**Summary:**

The paper proposes NP-Edit, a novel framework for training image editing diffusion models without any paired image data. Instead of relying on costly or synthetic edit supervision, NP-Edit trains a few-step diffusion model using direct gradient feedback from a Vision-Language Model (VLM), which evaluates whether an edit follows a natural language instruction while preserving unchanged content. In addition, the method incorporates Distribution Matching Distillation (DMD) to align edited outputs in the manifold of the pretrained T2I model. Experimental results show that NP-Edit achieves editing quality competitive with state-of-the-art methods..

**Strengths:**

S1. In the Abstract and Introduction section (Sec. 1.), the motivation for training an image editing model using the set of unpaired images are clearly explained. The paper is built on solid and strong motivation and requiredness of the task.

S2. The proposed components (loss functions) are plausible and details of VLM usage is clearly released in the Appendix. The idea of implementing unpaired training setup using the prior knowledge of VLM and manifold constraint is a novel approach.

S3. Related to S2, each loss function plays a critical role for performance improvement, as explained in the Ablation study section (Table 3). The paper clearly addresses the requiredness of each proposed loss term.

S4. The proposed method shows strong qualitative results against baselines, including image customization tasks. Quantitative results also reported, which shows the outstanding performance on GEdit-Bench dataset and on-part evaluation result on customization task.

**Weaknesses:**

W1. The method relies on the Vision Language Model (VLM) to judge whether the edited results of the model are correct. If the VLM misinterprets the image or instruction, the training signal becomes noisy, which can negatively impact convergence and editing accuracy. A more detailed analysis on VLM feedback dependency is required. How does the bias of VLM affect the model training procedure and overall performance? Is the Distribution Matching Distillation (DMD) enough to alleviate the bias of VLM?

W2. I was wondering if the method is applicable to more difficult tasks, such as 1) object addition or duplication, 2) object moving, and 3) enlarging or shrinking the object size. Extensive experiments with additional tasks is required.

W3. The method does not rely on supervision with paired before-to-after images. However, due to the absence of paired supervision, the edited output may drift from the reference object's identity, or introduce unwanted structural changes. Is there any analysis of the corresponding phenomenon? If the phenomenon that I mentioned is not crucial, please logically explain which part of the method alleviates this.

W4. Quantitative analysis on computation overhead is required. How much time is required to train the model and edit a single image?

**Questions:**

Please check the weakness section. It would also strengthen the paper to include a comparison with the recently released I2I model *Nanobanana*.

---

> ### Author Response · Authors · 2025-11-24
> **Response to reviewer anZe**
>
> Thank you for the positive comments regarding our motivation, methodology, and experimental validation. Below, we address the individual comments and questions:
>
> ### **Difficult tasks (W2) and VLM dependence (W1)**
> We agree that our method relies on the VLM’s capabilities and is subject to its biases. For example, while NP-Edit already successfully performs challenging edits such as object addition and shape changes (Appendix Figure 13, 2nd row and Figure 11 (a)), it fails at tasks requiring complex spatial reasoning (e.g., moving objects), as current VLMs struggle to provide meaningful feedback in such scenarios. We have added such sample failure cases in **Appendix Figure 11** (b).
>
> However, our framework is robust to sporadic VLM errors and optimizes the model based on the aggregate gradient signal over many iterations. We also observe better performance with larger backbones (Table 4), indicating our approach can benefit from future VLM advancements to address currently difficult tasks.
>
> ### **Drift between input and output edited image (W3)**
> We agree that, without pixel-level supervision, our method can drift from the input image, as discussed in our limitations (Lines 479-481) and Appendix C. We also demonstrated in Appendix C  that adding pixel-supervision losses, such as LPIPS, between the input and edited output image reduces drift but harms removal edits.
>
> Another effective approach to address this can be via cycle-consistency [1], as we show here. For an edited image (e.g., removing the flower pot), we apply a reverse edit (e.g., adding a flower pot) and encourage this reversal to reconstruct the original input image. We compute the reconstruction loss between the input and reverse-edited image in the SigLIP feature space. We select SigLIP’s global embedding over fine-grained feature spaces like LPIPS to provide flexibility when reverse instructions are ambiguous (e.g., restoring the exact original object). As shown in the updated **Figure 10 in Appendix C**, this helps reduce drift while preserving editability, though it still results in a drop in overall VIEScore (5.21 vs. 6.10). We have updated Appendix C with full details of the experiment.
>
> For local editing tasks specifically, another promising future direction can be to leverage masks, either user-provided or predicted, to strictly enforce pixel consistency in non-edited regions.
>
> We also observe that maintaining strict pixel-level consistency remains a challenge even for baselines trained on paired data, such as Qwen-Image-Edit (Figure 2, 1st row).
>
> [1] Zhu et al. Unpaired Image-to-Image Translation using Cycle-Consistent Adversarial Networks.
>
>
> ### **Computation requirement (W4)**
> **Training**: Our model training requires ~1600 GPU hours (32xA100 with 80GB VRAM for ~50h) at 512x512 resolution images. This is comparable to open-source baselines such as OmniGEN (trained on 104×A800 GPUs). We have also added this to our implementation details section in Appendix E.
> **Inference**: Our inference is quite efficient because of the few-step sampling: 4-step edit at 512x512 resolution takes 0.41 seconds on a single A100 GPU.
>
> ----
> **Comparison with Nano-Banana**: On the GEdit-Bench, it performs on par with the multi-step Qwen-Image-Edit baseline and outperforms all other methods, including ours. The semantic, quality, and overall VIEScore are 7.38, 7.86, and 7.05, respectively.
> While the compute, model architecture, and dataset requirements of Nano-Banana are unknown, it's likely to require large-scale paired data, R&D budgets, and engineering efforts, which are often challenging to obtain for a new task in the future. Our method aims to address this by proposing a paired-data-free training framework.

---

> > ### Comment · Reviewer_anZe · 2025-11-27
> >
> > I appreciate the authors' detailed response. I am maintaining my score as my concerns have been addressed. The challenges regarding task difficulty and reliance on VLMs have been discussed, and the cycle-consistency loss effectively reduces drift between the source and target images. The authors also clarified the required computational cost and explained the differences from the Nano-Banana model. Overall, the revisions sufficiently resolve the issues raised in my initial review.

---

### Public Comment · ~Bo_Zhao17 · 2025-11-14
**Question about your earth-shaking experiment result in Table 1**

I have read your paper. I wonder In Table 1 your method is huge better than qwen-edit in PQ ( image quality socre ) especially your model is only 2B. This reulst is unbelievable. In my personal opinion your method utilize MLLM, which may be excel in instruction following, not image quality.

---

> ### Author Response · Authors · 2025-11-14
> **Comparison is with original Qwen-Image-Edit model (w/o distillation)**
>
> Hi, thanks for your interest in our work. For clarification, in the few-step setting, we simply run the original Qwen-Image-Edit model, but with 4 steps, which results in low image quality. There is no distilled version of Qwen-Image-Edit that we are aware of.
>
> We also show the results with the full 50 steps of sampling in the first block of Table 1, where Qwen-Image-Edit performs on par with our method in image quality and has much better instruction following.

---

> > ### Public Comment · ~Jooyoung_Choi1 · 2025-11-17
> >
> > Hi authors, I enjoyed reading your paper. I just wanted to comment to let you know about the existence of timestep-distilled Qwen-Image-Edit: https://github.com/ModelTC/Qwen-Image-Lightning/. From my experience, its performance is good, and of course, I think your paper has high potential since it doesn't require paired data.
> > Applying your method to the timestep-distilled Qwen-Image-Edit to further improve its performance could be a good direction for future research.

---

> > > ### Author Response · Authors · 2025-11-24
> > > **Thanks for your comment**
> > >
> > > Hi,
> > >
> > > Thanks for bringing the Qwen-Image-Edit Lightning to our attention, and we will look into this model. We also agree that fine-tuning an existing distilled editing model using our framework is a promising direction that we aim to explore in future work.

---

### Author Response · Authors · 2025-11-24
**Author response summary**

We thank all the reviewers for their valuable feedback and detailed comments. We are excited that they found our method strongly motivated (anZe, jdS1, kKVu) with a novel (anZe, jdS1, kKVu, JFwR) and generalizable approach (kKVu) and rigorous evaluation (anZe, kKVu, JFwR).  Based on the comments, we have improved upon our draft and list the major changes below:

1. **Updated method diagram in Figure 1** (main paper)  to clarify our two-step sampling approach during training that predicts the clean edited image, which is then used for the VLM-based editing and DMD loss (jdS1).


2. **Additional evaluations in Appendix A**: human pair-wise comparison on GEdit-Bench under the few-step sampling regime (JFwR) and non-rigid edit types evaluation using TED-Bench (jdS1).
The overall trend remains similar to the VIEScore evaluation on GEdit-bench reported in Table 1. For the pair-wise human preference study, our method outperforms Turbo-Edit and remains competitive with other baselines, with a slight preference over FLUX.1-Kontext and Step-1X Edit and random chance with Qwen-Image-Edit, despite requiring much fewer parameters and no paired training data.

3. **Expanded Appendix C that discusses the current limitations** of our method regarding complex tasks and input-output drift (anZe, jdS1).
Though given the rapid pace of VLM advancement observed over the past few years, we anticipate that future VLMs, capable of more nuanced, pixel-level assessment, will help address these limitations.

We have **highlighted the new updates** and analyses, currently added to the Appendix, and are happy to incorporate any of them into the main paper according to the reviewers’ recommendations.

---

### Meta-Review · Area_Chair_nAt9 · 2025-12-02

**Summary:**

This paper proposed a training method for image editing diffusion models. The proposed method aims to mitigate the reliance on paired images which are requested by contemporary in-context DiT model.  Leveraging differentiable feedback from the VLM, the proposed method optimizes a few-step diffusion generator based on a reference image and an instruction. In the few-step setting, it achieves competitive performance.

Three reviewers give positive rating scores and one reviewer give negative score. According to the authors' rebuttal, they have provided a comprehensive response to the reviewers' concerns and have high expectations for an improvement in the reviewers' scores. Based on the above considerations, I recommend to accept this manuscript.

**Reviewer Concerns:**

Multiple reviewers (Reviewer anZe, jdS1, kKVu, JFwR) raised concerns about the impact of inaccurate VLM feedback on the performance of the proposed method. The reviewers provide corresponding analyses on limitations and sensitivity to VLM's feedback. Additionally, authors conduct more validation on non-rigid edit types, and optimize the presentation about obtaining the editing result fed to VLM during training,  to address Reviewer jdS1's concerns.

**Reviewer Scores:**

Reviewer anZe, kKVu, JFwR may maintain their positive rating score and Reviewer jdS1 may change the rating score to a positive one.

---

### Decision · Program_Chairs · 2026-01-26

Accept (Poster)